# Fast and sensitive flow-injection mass spectrometry metabolomics by analyzing sample-specific ion distributions

Boris Sarvin [1,4], Shoval Lagziel [2,4], Nikita Sarvin[1], Dzmitry Mukha[1], Praveen Kumar[1], Elina Aizenshtein[3] & Tomer Shlomi [1,2,3 ✉]

Mass spectrometry based metabolomics is a widely used approach in biomedical research. However, current methods coupling mass spectrometry with chromatography are time-consuming and not suitable for high-throughput analysis of thousands of samples. An alternative approach is flow-injection mass spectrometry (FI-MS) in which samples are directly injected to the ionization source. Here, we show that the sensitivity of Orbitrap FI-MS metabolomics methods is limited by ion competition effect. We describe an approach for overcoming this effect by analyzing the distribution of ion *m/z* values and computationally determining a series of optimal scan ranges. This enables reproducible detection of ~9,000 and ~10,000 *m/z* features in metabolomics and lipidomics analysis of serum samples, respectively, with a sample scan time of ~15 s and duty time of ~30 s; a ~50% increase versus current spectral-stitching FI-MS. This approach facilitates high-throughput metabolomics for a variety of applications, including biomarker discovery and functional genomics screens.

[1] Faculty of Biology, Technion—Israel Institute of Technology, 32000 Haifa, Israel. [2] Faculty of Computer Science, Technion—Israel Institute of Technology, 32000 Haifa, Israel. [3] Lokey Center for Life Science and Engineering, Technion—Israel Institute of Technology, 32000 Haifa, Israel. [4] These authors contributed equally: Boris Sarvin, Shoval Lagziel. ✉email: tomersh@cs.technion.ac.il

Metabolomics and lipidomics enable the detection of numerous molecules in biological samples and are extensively used to explore the dynamic response of living systems to diverse physiological and pathological conditions[1,2]. Untargeted metabolomics and lipidomics are especially important for unbiased detection of a wide range of molecules, providing an important tool for new discoveries in metabolic studies[3–8].

Metabolomics is typically performed via nuclear magnetic resonance and mass spectrometry (MS)[9,10]. MS is typically coupled with liquid or gas chromatography, separating metabolites within a complex sample before ion detection[11,12]. This enables separating isobaric compounds, determine structural information, and further increases the overall sensitivity of the analysis by minimizing the effect of ion suppression[12,13]. However, the major limitation of this approach for metabolomics analysis is that chromatographic separation is time-consuming (typically requiring 20–60 min per sample), preventing its usage for high-throughput metabolomics and lipidomics screens of thousands of samples—required for large scale biomarker discovery studies and functional genomics screens[12,14]. Ultrashort columns provide shorter chromatographic separation times (5 min or less), though are typically limited in terms of the capability to separate complex mixtures.

An alternative analytical approach is flow-injection mass spectrometry (FI-MS), in which the analytes are directly injected into the mass spectrometer ionization source without prior chromatographic separation. FI-MS via modern high-resolution mass spectrometers such as Orbitrap and time-of-flight (ToF) was shown to enable the determination of hundreds to thousands of m/z features in biological samples[15–17]. FI-MS based on Orbitrap and ToF provide complementary analytical capabilities, considering the inherent differences between the two mass spectrometers; e.g., with Orbitrap providing higher resolution for low mass ions and ToF for high mass ions[18]. FI-MS based on ToF MS was shown to enable metabolomics analysis time of ~1 min per sample[16]. This was recently applied to perform metabolomics screens in cancer cell lines and study the metabolic response to drug treatment in cancer cells and bacteria[19–23]. Recently, a FI-MS method using a high-resolution Orbitrap mass spectrometer was proposed, enabling metabolomics and lipidomics analysis of ~9000 m/z features in ~5 min per sample[17]. This method, referred to as spectral stitching, aims to minimize the ion overload of the mass spectrometer by configuring a quadrupole to separately pass ions within consecutive m/z intervals to the Orbitrap analyzer. This increased the overall sensitivity fivefold, compared with a naïve approach in which the entire m/z range of interest is scanned at once[17,24]. However, splitting the mass spectrometer scanning to a large number of small ranges increased the overall scanning time to several minutes—limiting the applicability of this approach for high-throughput screening[17].

Here, we present a method for improving the sensitivity of rapid FI-MS with a high-resolution Orbitrap mass spectrometer, determining the optimal scan ranges that would maximize the number of reproducibly detected metabolites and lipids. The method is based on first running FI-MS with consecutive and narrow scan ranges to inspect the distribution of reproducibly detected m/z features within an m/z scan interval of interest. These measurements are then used to divide that m/z scan interval of interest into a small set of scan ranges that would maximize the overall sensitivity (aiming to achieve a uniform number of reproducibly detected m/z features in each scan range). We demonstrate the applicability of this approach in untargeted analysis of metabolites and lipids in serum samples as well as of metabolites extracted from cancer cells grown in tissue culture. We explore the trade-off between analysis time and sensitivity, analyzing how the number of analyzed scan ranges determines the number of reproducibly detected m/z features. For example, we show that with eight optimized scan ranges, the metabolomics and lipidomics analysis (each with a scan time of 15 s) is sufficient to reproducibly detect a total of ~19,000 m/z features; ~50% higher than the number of features detected with a similar number of uniform size scan ranges. We expect this approach, facilitating rapid and high-sensitivity FI-MS analysis, to be highly useful for high-throughput metabolomics and lipidomics applications; e.g., for population-level disease screening and functional genomics screens aiming to assess gene metabolic activities and drug effects.

## Results

**Ion competition lowers the sensitivity of FI-MS metabolomics.** FI-MS involves simultaneous infusion of numerous ions to the mass spectrometer which typically lowers the sensitivity of the measurement (as the presence of highly abundant ions mask the detection of other lowly abundant ions). This may involve two distinct effects: (1) ion suppression at the ionization source (electrospray ionization (ESI) source, in our case). (2) Ion competition in the detection system, typically having a limited capacity in terms of the number of accumulated ions; in our case, a curved linear trap (C-trap), capturing up to $5 \times 10^6$ ions prior to their transfer to the Orbitrap analyser (see "Methods"). Ion competition in the detection system can be lowered by configuring a quadrupole to limit the range of ion m/z that are transferred to the Orbitrap analyser at a given time, as performed with the spectral-stitching approach[17]. Here, we aimed to assess the extent of ion suppression in the ESI source versus ion competition in the detection system, as a mean to evaluate the potential of spectral stitching to improve the sensitivity of FI-MS analysis.

Toward this end, we utilized FI-MS with a high-resolution Hybrid Quadrupole Orbitrap mass spectrometer with an ESI source and performed the following experiment: we injected a series of serum samples in which the ion flow was gradually induced by adding increasing concentrations of a highly ionizable compound, taurocholic acid (TC), $[M - H]^-$—514.28 m/z; from 10 to 250 μM (Supplementary Fig. 1a; "Methods"). We configured two ranges in the quadrupole: a 24 m/z scan range that spans this compound, and a smaller 20 m/z scan range (enclosed within the previous scan range) in which the m/z signal of TC is excluded (Fig. 1a; "Methods"). The effect of ion suppression in the ESI was assessed based on the drop in the number of reproducibly detected m/z features in the 20 m/z scan range (which excludes m/z signal of TC) when adding TC (see definition of reproducible m/z features in "Methods"); showing a drop of no more than 25% when adding a maximal concentration of 250 μM TC (blue line, Fig. 1b). The total effect of ion competition in the detection system and ion suppression in ionization source together was assessed based on the drop in the number of detected m/z features when configuring a 24 m/z range (that includes the m/z signal of TC) when adding TC (considering only on m/z features within the smaller 20 m/z interval, without TC). This showed a markedly larger effect of ion competition in the detection system than of ion suppression in the ESI, with a drop of up to 90% in the number of reproducibly detected m/z features when adding the maximal concentration of TC (red line, Fig. 1b).

The overloading of the detection system with TC is only observed when the quadrupole is configured with the wider 24 m/z scan range that spans TC based on a drop in the ion injection (accumulation) time within the C-trap when adding increasing concentrations of TC (Fig. 1c); controlling the injection time enables the MS to prevent overloading of the Orbitrap analyser, while the automatic gain control (AGC) limits the maximal

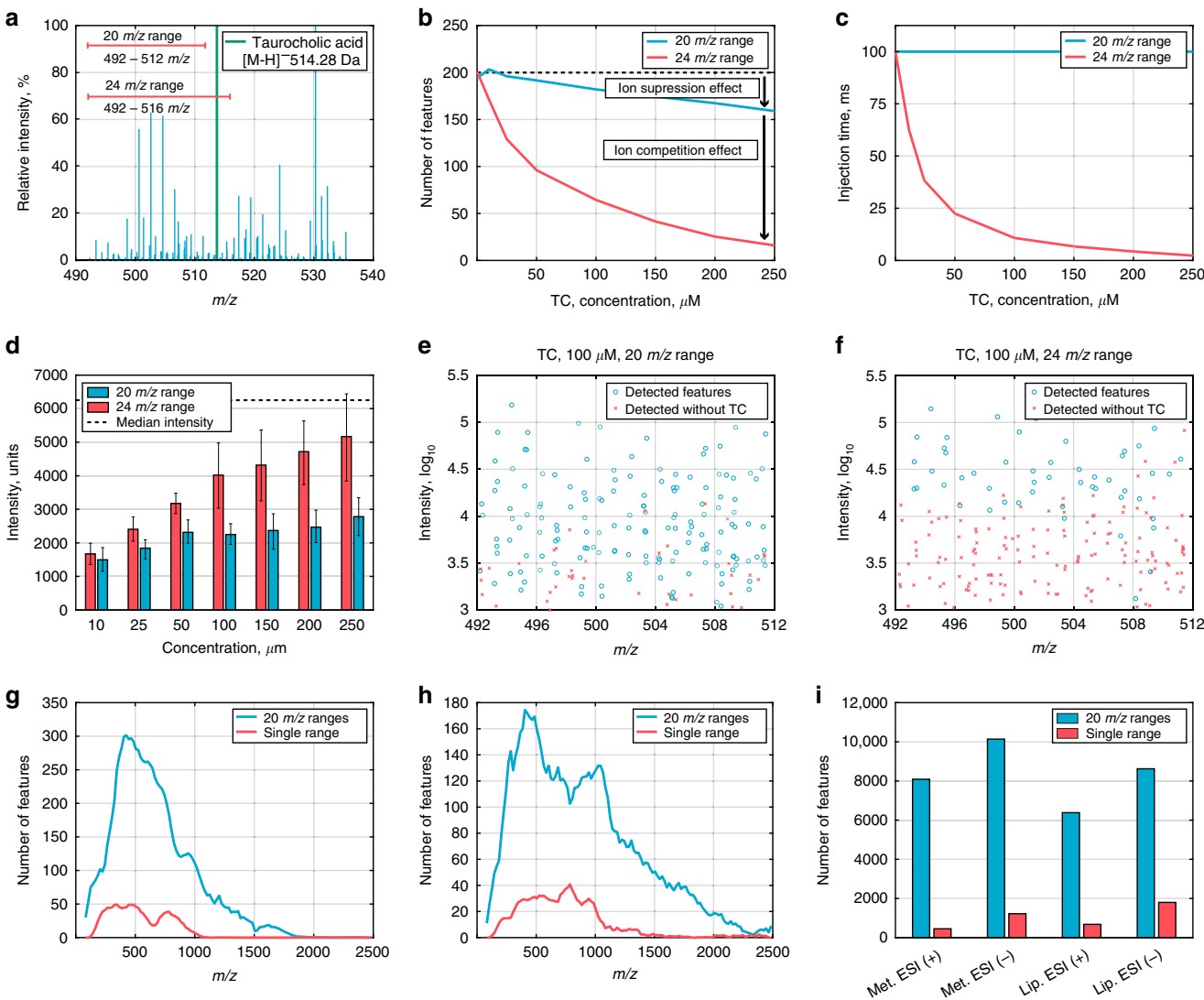

**Fig. 1 Ion competition in the detection system explains the reduced sensitivity of FI-MS. a** An experimental scheme for investigating ion suppression and ion competition effects in FI-MS analysis: gradually increasing the ion flow by adding increasing concentrations of some compound to the analyte, while configuring the mass spectrometer to scan for two overlapping ranges that include or exclude the added compound; here, taurocholic acid (TC) was added to metabolite extracts from serum samples, while a 20 *m/z* scan range, which excludes this compound and an overlapping 24 *m/z* scan range that includes it are scanned. **b** The number of reproducible *m/z* features found (within the narrower scan range) when scanning for the 20 *m/z* scan range (in blue) and for the 24 *m/z* scan range (in red), adding increasing concentrations of TC; black horizontal line represents the number of *m/z* features detected without adding TC. **c** Observed injection time of curved linear trap as a function of the concentration of the added TC. **d** The median intensity of *m/z* features ($n \geq$ 15; ±SEM; measured without adding TC; *y* axis) which become undetected when adding increasing concentrations of TC (*x* axis), scanning for the 20 *m/z* scan range (in blue) and for the 24 *m/z* scan range (in red); black horizontal line represents the median intensity of *m/z* features detected without adding TC. Ion intensity (*y* axis) and *m/z* (*x* axis) detected in serum when scanning for the 20 *m/z* scan range (**e**) and the 24 *m/z* scan range (**f**); ions undetected when adding 100 μM of TC are marked with red crosses. Distributions of the number of reproducible *m/z* features found by spectral-stitching FI-MS method (20 *m/z* scan ranges; in blue) versus scanning using a single range (in red) for metabolomics (**g**) and lipidomics (**h**) analysis (negative ionization mode). **i** The total number of reproducible *m/z* features found by spectral-stitching FI-MS (in blue) versus scanning using a single scan range (in red), for metabolomics and lipidomics, in positive and negative ionization modes. Source data are provided as a Source data file.

number of ions that will be transferred to the detection system (in our case configured to a maximal value of $5 \times 10^6$ ions). When the quadrupole is configured with the narrower 20 *m/z* scan range that excludes TC, a maximal ion accumulation time of 100 ms was realized (in accordance with our configuration of the MS method). Expectedly, the ion competition effect observed when scanning for the 24 *m/z* range spanning TC masks the detection of low-abundant and/or poorly ionized compounds having low intensity (Fig. 1d–f); gradually increasing the concentration of TC leads to the loss of *m/z* features having higher and higher intensities (Fig. 1d). The markedly larger effect of ion competition in the

detection system rather than ion suppression in the ESI on the number of reproducibly detected *m/z* features was further found when adding other compounds (sodium dodecyl sulfate (SDS), caffeine, Met-Arg-Phe-Ala (MRFA) peptide, and β-nicotinamide adenine dinucleotide (NAD)), and when configuring the quadrupole for passing ions within different scan ranges (Supplementary Figs. 1–3). Overall, our results suggest that ion competition in the detection system is the prime reason for the drop in sensitivity in Orbitrap-based FI-MS.

Ion competition in the detection system can be addressed by configuring the quadrupole to limit the scan range of *m/z* features

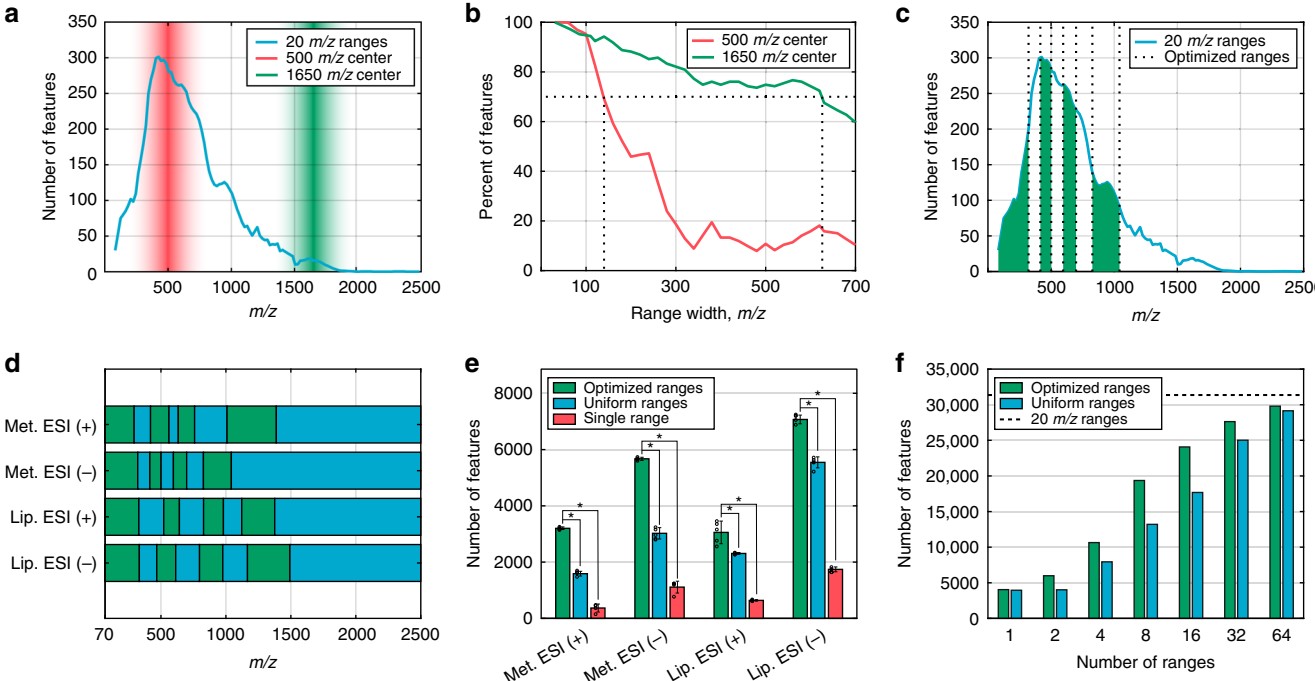

**Fig. 2 Optimizing the sensitivity of rapid flow-injection mass spectrometry via nonuniform scan ranges. a** An experimental scheme for investigating the effect of scan range width on the number of reproducibly detected $m/z$ features: FI-MS analysis of metabolite extracts from serum samples is performed with a series of scan ranges with increasing sizes around a point of 500 $m/z$ (where the mass-stitching revealed numerous reproducible features; in red) and a point of 1650 $m/z$ (where a small number of features are observed; in green). Specifically, using a series of ranges of size 40–700 $m/z$, in steps of 40 $m/z$ units. **b** The percent of reproducible $m/z$ features identified around the points of 500 (in red) and 1650 $m/z$ (in green), as a function of the scan range width (out of the total number of features with the minimal 40 $m/z$ scan range). **c** The distribution of the number of reproducible features found for metabolomics analysis of serum samples by FI-MS in negative mode; and the optimized eight scan ranges represented by vertical lines. **d** The optimized scan ranges for metabolomics and lipidomics FI-MS-based analysis of serum samples in positive and negative ionization modes. **e** The number of reproducible $m/z$ features identified with the optimized eight scan ranges (in green), eight uniform scan ranges (in blue), and using a single scan range (in red). *$p < 0.004$ by two-sample two-sided $t$-test. Data are mean ± SD, $n = 5$ independent repetitions of the FI-MS analysis. **f** The total number of reproducible $m/z$ features ($y$ axis; metabolomics and lipidomics analysis via positive and negative ionization modes) identified when scanning using optimized scan ranges (in green) versus when using uniform width scan ranges (in blue), considering different numbers of scan ranges ($x$ axis); the black horizontal line marks the number of reproducible features observed when scanning using the exhaustive 20 $m/z$ scan ranges. Source data are provided as a Source data file.

that are transferred to the C-Trap at any given time. Accordingly, we set up spectral-stitching FI-MS-based metabolomics and lipidomics methods for the analysis of serum samples, scanning an $m/z$ interval of 70–2500, via a series of 122 consecutive 20 $m/z$ scan ranges (involving multiple injections of each sample to scan a subset of these $m/z$ ranges; see "Methods"). The 70–2500 $m/z$ interval was chosen considering that the vast majority of serum metabolites and lipids are within this mass range, according to the Human Metabolome Database (HMDB)[25] and LIPID MAPS Structure Database (LMSD)[26] (Supplementary Fig. 4). The metabolomics method detects 6815 and 8713 $m/z$ features in positive and negative ionization modes, respectively, while the lipidomics method detects 6695 and 9138 $m/z$ features in the positive and negative ionization modes, respectively (Fig. 1g–i). This represents a marked improvement in overall sensitivity compared with scanning a single range between 70 and 2500 $m/z$; with approximately tenfold increase in the number of reproducible $m/z$ features for the metabolomics analysis and approximately sixfold increase for the lipidomic analysis (Fig. 1i). However, a major limitation of this approach is that the high number of scanned ranges result in a long running time which is on the order of minutes (considering a scan rate of 1.3 Hz at a high-resolution of 70,000 and the requirement of 3 micro scans to achieve stable and reproducible measurements; see "Methods").

**$m/z$ ranges optimization improves the sensitivity of FI-MS.** We hypothesized that controlling scan range widths would provide the needed flexibility to minimize the number of scan ranges while maximizing the overall number of detected $m/z$ features. To explore how the width of the scanned range affects the resulting number of detected features, we performed the following experiment: we configured FI-MS method to scan for series of ranges with increasing size, centered around a point of 500 $m/z$ (a high-density spectral region where the above mass-stitching analysis revealed numerous reproducible features) and around a point of 1650 $m/z$ (a low-density spectral region where a significantly lower number of features were found; Fig. 2a). Specifically, around each of the center $m/z$ points, we scan a series of ranges of size 40 $m/z$ to 700 $m/z$, in steps of 40 $m/z$ units. Expectedly, for both scans performed within high- and low-density spectral regions, the number of detected $m/z$ features gradually dropped when increasing the scanning range widths due to ion competition effect in the detection system described above (compared with the number of features detected with a 40 $m/z$ scan range, Fig. 2b). However, the drop in the number of reproducibly detected $m/z$ features was markedly larger around the more dense region of 500 $m/z$ versus the low-density region around 1650 $m/z$. Specifically, to detect more than 70% of the features found using the 40 $m/z$ range, a maximal scan range size of ~140 $m/z$ could be used for high-density spectral region; versus

a maximal range size of ~620 *m/z* for low-density spectral region. Overall, these results demonstrate that low-density spectral regions tolerate wider scan range, but in mass ranges with a high ion density dense sampling is crucial.

Considering the above, we developed a method to determine the optimal set of scan ranges that aims to maximize the total number of reproducibly detected *m/z* features by achieving a uniform number of *m/z* features in each scan range. The method takes as input the distribution of the number of reproducibly detected *m/z* features within a certain *m/z* interval (as determined via an exhaustive spectral-stitching method with small and uniform scan ranges described above) as well as the number of requested scan ranges (determining the overall FI-MS analysis time). It then finds the set of scan ranges that achieve a uniform number of reproducible *m/z* features in each scan range. For example, applying this method to determine an optimal set of eight scan ranges for metabolomics analysis of serum samples with negative ionization mode identified the ranges shown in Fig. 2c (where the number of reproducible *m/z* features within each scan range is equal). Performing the analysis also for metabolomics analysis in positive ionization mode resulted in different optimal scan ranges; and similarly, applying it for lipidomics analysis resulted in different scan ranges (Fig. 2d, Supplementary Fig. 5). Notably, applying this method to several serum samples from different donors (and to several different mice brain samples) showed that different samples of the same type produce a similar characteristic distribution of *m/z* features (Supplementary Fig. 6); and hence scan range optimization should be performed once for every sample type of interest.

The scan time for metabolomics or lipidomics analysis with eight scan ranges in both positive and negative ionization required a total of ~15 s (duty time of ~30 s; time between two consecutive injections). The eight optimal scan ranges for metabolomics and lipidomics analysis enabled the detection of a total of ~19,000 *m/z* features in serum samples. This represents a significant approximately fivefold increase in the number of reproducible features versus utilizing a single scan range (*p* value $< 10^{-10}$, two-sample *t*-test) and a significant ~50% increase compared with utilizing eight uniform scan ranges (*p* value $< 10^{-7}$; Fig. 2e). Overall, the metabolomics and lipidomics analysis with eight scan ranges identify 60% of the *m/z* features detectable by the exhaustive 20 *m/z* scan ranges spectral-stitching (~5 min of scanning time). As an alternative approach for determining the scan ranges, we tested a method that aims to achieve a uniform total ion count (TIC) within each range, though this leads to a significantly lower overall number of reproducible *m/z* features (Supplementary Fig. 7).

We comprehensively evaluated the trade-off between the number of scan ranges and the number of reproducibly detected *m/z* features with our approach (Fig. 2f). We found that 4 scan ranges are sufficient to detect ~1/3 of the total number of reproducible *m/z* features identified with the exhaustive 20 *m/z* scan ranges spectral-stitching FI-MS experiment; on the other hand, 16 scan ranges are sufficient to identify >75% of the reproducible *m/z* features detectable with the exhaustive analysis. Overall, the advantage of the optimized scan ranges is evident across a wide number of potential scan ranges.

We further evaluated the analytical performance of the FI-MS method in terms of sample-specific matrix effect that may bias metabolite concentration measurements and linearity of response to changes in metabolite composition. Sample-specific matrix effect was evaluated by applying the optimized eight scan ranges FI-MS-based metabolomics and lipidomics methods to analyze 20 serum samples from different donors while injecting internal standards (glucose 6-phosphate (G6P), $[M - H]^{-}$—259.02 *m/z*; and MRFA, $[M + H]^{+}$—524.26 *m/z*; Fig. 3a). We injected 250 nM of G6P and 150 nM of MRFA, such that their measured intensity

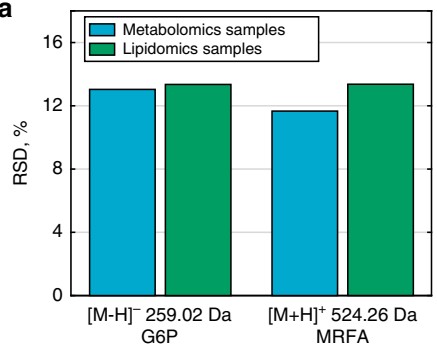

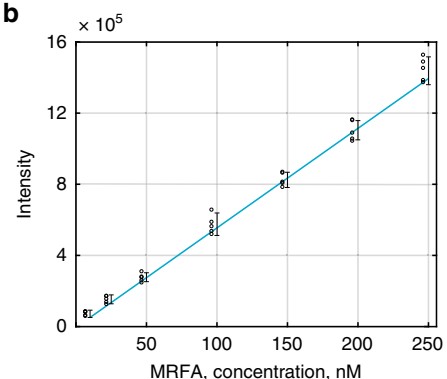

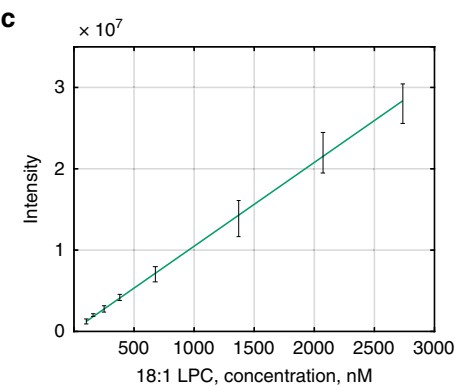

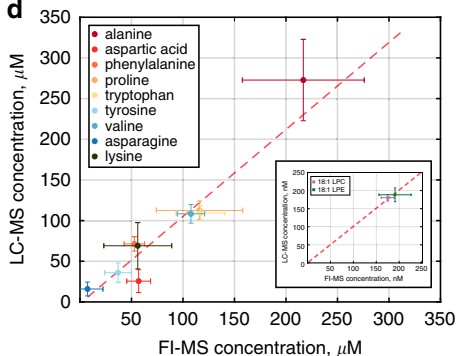

matches the median intensity of all reproducibly detected *m/z* features in these samples. For both internal standards in metabolomics and lipidomics, the relative standard deviation (RSD) in the measured intensity across the 20 serum samples was <14%, supporting the ability of our method to overcome sample-specific matrix effect and correctly detect that the same concentration of these compounds was injected to all samples. To examine the linearity of the measurements, we applied

**Fig. 3 Sample-specific matrix effect and a linear response observed with optimized FI-MS. a** Applying FI-MS method with 8 optimized ranges in negative ionization mode for metabolomics and lipidomics analysis of serum samples from 20 donors and adding 250 nM of glucose 6-phosphate (G6P); and in positive ionization mode, adding 150 nM of Met-Arg-Phe-Ala tetrapeptide (MRFA); $y$ axis represents the RSD% of the measured intensity of these compounds across the 20 analyzed samples. Applying FI-MS with eight optimized ranges in positive ionization mode for metabolomics analysis of serum samples, adding increasing concentrations of taurocholic acid (**b**) and lipidomics analysis in positive ionization mode adding increasing concentrations of 18:1 lyso phosphocholine (LPC) (**c**), demonstrating a linear response in both cases. Data are mean values ± SD, $n = 5$ (10 for LPC measurement) independent repetitions of the FI-MS analysis. **d** Correlation between FI-MS and LC-MS-based concentration measurements of nine amino acids and two lipids in serum. Error bars show 95% confidence intervals. Source data are provided as a Source data file.

metabolomics based FI-MS to analyze a serum sample injected with a series of increasing concentrations of TC, and lipidomics based FI-MS for serum samples injected with 18:1 lyso phosphocholine. In both cases, the calibration curves showed a highly significant linear response over a concentration range spanning two orders of magnitude ($R^2 > 0.99$; $p$ value < $10^{-5}$; Fig. 3b, c). A Similar linear response spanning a concentration range above two orders of magnitude was found for nine other injected standards in metabolomics and lipidomics analysis (Supplementary Table 1; "Methods"), supporting the ability of our method to quantify relative changes in metabolite abundances between samples. To test the ability of our FI-MS method to quantify the absolute concentration of metabolites, we utilized the standard addition method to determine the concentration of nine amino acids; and utilized internal calibration curves with stable isotope labeled standards to determine the concentration of two lipids in serum. The inferred concentrations by our FI-MS method are significantly correlated with those determined via standard addition performed with LC-MS (Pearson $r = 0.95$, $p$ value < $10^{-5}$, Fig. 3d; Supplementary Table 2; "Methods").

We obtained putative annotations for $m/z$ features detected with our FI-MS method, based on the high-accuracy Orbitrap MS measurements, compared with HMDB and LMSD (see "Methods"; level 2 annotations[27]). For example, several metabolite classes which are known to be present in blood were annotated, including lipids (fatty acids and esters, phospholipids, sterols, and ceramides) and polar compounds (amino acids, pyrimidine nucleosides, peptides, and carbohydrates). The number of $m/z$ features detected with our optimized ranges FI-MS method with a putative annotation is significantly higher (approximately twofold increase) than for the set of $m/z$ features detected with uniform range FI-MS (two-sample $t$-test $p$ value < $10^{-9}$); and is significantly higher (~3.5-fold increase) than for the set of $m/z$ features detected with FI-MS with a single rage ($p$ value < $10^{-9}$; Supplementary Fig. 8). Notably, more than 30 fatty acids, 20 steroids, and 15 carbohydrates are nondetectable with uniform scan ranges but could be detected using the optimized FI-MS method (Supplementary data 1).

We applied our FI-MS method to analyze inter-subject variability in serum metabolome within a group of 98 healthy individuals ("Methods"). Our analysis shows variability in the abundance of ~3500 $m/z$ features across the analyzed population (i.e., number of $m/z$ features whose inter-subject RSD is 50% higher than the RSD in repeated injection of QC samples[28]); this number of biologically important $m/z$ features is only ~20% lower than that detected in a recent study via standard LC-MS analysis, though the FI-MS analysis is ~100-fold faster[28]. We find that the distribution of inter-subject RSD is skewed to lower values, in accordance with the LC-MS-based results (Supplementary Fig. 9; Supplementary data 2)[28]; with a similar median inter-subject RSD of ~40% and maximum RSD of ~700%. Specifically, in accordance with the LC-MS results, we find that aromatic amino acids (tryptophan, phenylalanine, and tyrosine) have low inter-subject RSD (<~35%); fatty acids (tetradecanoic and hexadecenoic acids) have intermediate RSD values (66 and 71%, respectively); and, expectedly, drug metabolites (paracetamol, sitaxentan) have the highest RSD (>300%). Utilizing the entire set of serum FI-MS measurements to reproduce a study of gender-specific metabolite fingerprints performed with LC-MS, resulted in overall similar results (Supplementary Fig. 10; "Methods").

**FI-MS metabolomics analysis of cancer cells**. We applied our method to determine the optimal scan ranges for FI-MS analysis of metabolites from cultured cancer cells: first, we applied FI-MS-based metabolomics in positive and negative mode with exhaustive 20 $m/z$ scan ranges on metabolite extracts from cultured HeLa cells and from the cell culture media. This was used to derive distributions of the number of reproducible $m/z$ features within the $m/z$ interval of 70–2500 $m/z$ for intracellular and media metabolites with positive and negative ionization mode. These distributions were utilized to determine sets of eight optimal scan ranges for the metabolomics analysis in each ionization mode (Fig. 4a). The resulting FI-MS-based metabolomics method detects a total of ~3800 and ~7200 $m/z$ features in cells and media extracts, respectively (in both polarity modes combined). This represents a major improvement compared with utilizing uniform scan ranges; with a significant approximately twofold and ~2.5-fold increase in the number of $m/z$ features detected in positive ionization mode with our optimized ranges versus with uniform ranges for cell and media extracts, respectively ($p$ value < $10^{-11}$ and < $10^{-7}$; Fig. 4b, c). More than 1300 $m/z$ features were putatively annotated based on the accurate mass measurement (using HMDB; level 2 annotations[27]), including purines, pyrimidines, amino acids, and sterols (Supplementary data 3).

To evaluate the performance of our FI-MS method in accurately determining metabolite abundances across cancer cells, we systematically applied it to analyze metabolite extracts from ten cell lines (HeLa, Hek293, HepG2, MiaPaca2, HCT116, Panc-1, A549, WM266-4, Jurkat, and CCRF-CEM) and compared the results to those obtained with untargeted LC-MS analysis ("Methods"). We found a total of 815 $m/z$ features that are identified by both LC-MS and FI-MS within at least seven cell lines, in positive and negative ionization modes combined (utilizing MAVEN[29] to extract reproducibly intense MS peaks with intensity > 15,000 in negative and positive ionization modes). Ion intensity measurements performed by our FI-MS method across the cell lines are significantly correlated with those made by LC-MS for a total of 367 $m/z$ features in negative and positive modes (FDR corrected Pearson $p < 0.05$; Supplementary data 4). For FI-MS with uniform ranges, a significant correlation with LC-MS measurements was obtained for only 216 $m/z$ features; and for FI-MS with a single range for only 86 $m/z$ features (Supplementary Fig. 11). Furthermore, the correlations between LC-MS intensity measurements and those made via our optimized ranges FI-MS are significantly higher than with those made with FI-MS with uniform ranges and with FI-MS with a single range (Wilcoxon rank-sum $p$ value < $10^{-3}$ and < $10^{-7}$, respectively). For example, for uridine diphosphate $N$-acetylglucosamine, the correlation between measurements performed with our optimized ranges FI-MS (negative ionization mode) and LC-MS ($r = 0.95$, FDR corrected $p$ value < $10^{-6}$) is markedly higher

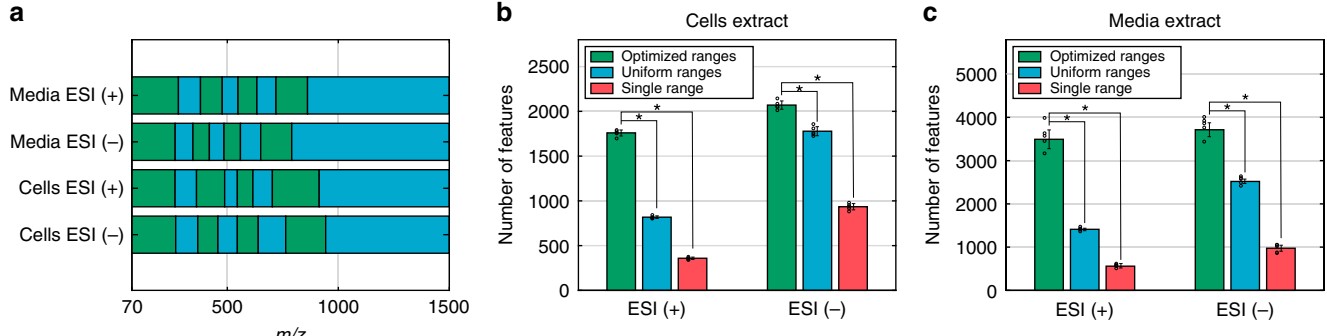

**Fig. 4 FI-MS-based metabolomics with optimized scan ranges applied for cultured cancer cells. a** The optimized eight scan ranges for metabolomics FI-MS-based analysis of metabolite extracts from cancer cells and from culture media in positive and negative ionization modes. The number of reproducible *m/z* features observed when utilizing the eight optimized scan ranges (in green), compared with eight uniform scan ranges (in blue), and a single scan range (in red), analyzing metabolite extracts from cancer cell (**b**) and from culture medium (**c**). *$p < 10^{-5}$ by two-sample two-sided *t*-test. Data are mean values ± SD, n = 5 independent repetitions of the FI-MS analysis. Source data are provided as a Source data file.

than that for uniform range FI-MS ($r = 0.87$, FDR corrected *p* value < 0.05) and single range FI-MS ($r = 0.78$, FDR corrected *p* value > 0.05).

As a model system for further testing whether our FI-MS method could capture important biological alterations in cellular metabolome, we reproduced a study on the effect of oxygen level on intracellular metabolites performed via LC-MS[30]. We applied the developed FI-MS method to measure the metabolome of HCT116 cancer cells grown in hypoxia (1% $O_2$, see "Methods") and normoxia (20% $O_2$). We find that for a set of 31 metabolites whose concentration was reported to increase under hypoxia based on LC-MS, our FI-MS analysis also shows a significant increase in abundance in hypoxia (Wilcoxon rank-sum *p* value < $10^{-7}$; comparing the fold change in the abundance of these metabolites in hypoxia versus hypoxia with that of other detectable metabolites; Supplementary Table 3).

## Discussion
We presented an approach for optimizing Orbitrap FI-MS analysis of specific samples for both high speed and sensitivity—making FI-MS suitable for high-throughput metabolomics and lipidomics applications. We showed that while both ion competition and ion suppression hinder the sensitivity of FI-MS-based metabolomics and lipidomics analysis, ion competition is the predominant factor. To overcome the effect of ion competition in the mass spectrometer detection system while minimizing the overall scan time, we developed a method for identifying a predefined number of *m/z* scan ranges that would maximize the overall sensitivity for a specific sample type of interest—aiming to achieve a uniform number of reproducibly detected *m/z* features in each *m/z* scan range (i.e., practically assigning narrow ranges around *m/z* regions having a high number of features and wider ranges around sparse *m/z* regions). This was shown to increase the number of reproducibly detected *m/z* features by ~50% and >100% in metabolite extracts from serum and cultured cells, respectively, compared with spectral-stitching FI-MS with uniform *m/z* scan ranges.

Reproducibly detected *m/z* features by FI-MS do not necessarily reflect distinct metabolites in the analyte, considering the presence of adducts, in source fragments, and natural isotopes. Several methods were proposed to group multiple ions detected with LC-MS based on a likely association with the same metabolite in the analyte. For example, clustering of *m/z* peaks based on high correlation between peak shapes (i.e., time-dependent intensity measurement throughout the chromatographic separation)[31]. Analogously, we performed clustering of *m/z* features

detected with FI-MS based on a high correlation between their measured intensity across a series of different analyzed samples ("Methods"). Applied to cluster *m/z* features identified by our optimized ranges FI-MS method across a series of 98 serum samples, we obtained a total of ~3900 *m/z* feature clusters (Supplementary Fig. 12). A markedly lower number of ~1200 *m/z* clusters was obtained by the uniform ranges FI-MS and only ~250 clusters with a single range FI-MS, further supporting the higher metabolite coverage of our optimized ranges FI-MS method.

Metabolite annotation is a major challenge in both untargeted LC-MS and FI-MS-based metabolomics and lipidomics. Here, we obtained putative annotations for identified *m/z* peaks based on the high mass accuracy of the Orbitrap mass spectrometer (compared with public metabolite databases); level 2 annotations, as defined by the MSI[27]. We showed that the number of detected *m/z* features for which a putative annotation could be obtained is significantly higher for measurements performed with our optimized ranges FI-MS method than with FI-MS with uniform ranges or a single range. Obtaining higher confidence annotations for *m/z* features detected by FI-MS is possible via MS/MS analysis[32,33]. Applying FI-MS/MS with data-dependent analysis (DDA) to induce collisional fragments for high intensity ions, we derive high confidence (level 1) annotation for 83 *m/z* features, including 22 polar compounds annotated based on METLIN MS² spectra and 61 nonpolar compounds annotated based on LMSD (Supplementary data 5). For many of the latter, FI-MS enables the annotation of lipid class, total length of acyl chains, and total number of unsaturated bonds, while more specific annotation would require an extra level of separation beyond *m/z*. Numerous additional *m/z* features detected by FI-MS could be annotated with parallel reaction monitoring, systematically utilized to acquire MS/MS spectra for *m/z* features of interest.

We show that for different sample types (serum, mice brain extracts, intracellular and media metabolites) and analysis type (metabolomics and lipidomics; positive and negative ionization mode), our method finds different sets of optimal *m/z* scan ranges, suggesting that it would be highly useful to utilize this approach to determine optimal *m/z* ranges for future applications of FI-MS-based metabolomics and lipidomics. Overall, our FI-MS method is expected to be an important tool for metabolomics and lipidomics based functional genomics and biomarker discovery studies.

## Methods
**Chemicals**. Water, methanol, acetonitrile, and 2-propanol—LiChrosolv LC-MS grade Merck & Co. (Germany), buffer additives ammonium carbonate (for HPLC) and ammonia solution, 25% (for LC-MS) were purchased from Fluka Analytical

Sigma-Aldrich (Germany) and Merck & Co. (Germany), respectively. Compounds for investigation of ion competition and suppression effects and MS $m/z$ accuracy were received from Sigma-Aldrich (Germany). Amino acids and stable isotope labeled lipids standards were obtained from Biological Industries, Inc. (USA) and Avanti Polar Lipids, Inc. (USA), correspondingly.

**Biological Samples**. FI-MS method optimization was performed with commercially available serum, Human AB Serum (Biological Industries USA, Inc., USA). Matrix effect experiments and biological applications were performed with 98 serum samples of healthy individuals obtained from Rambam Hospital, Haifa, Israel (Supplementary Table 4). All serum samples of healthy individuals were purchased from Midgam Biobank during December 2018. All participants provided their informed consent and the study protocol was approved by the Ethics Committee of Rambam Health Care Campus, Technion—Israel Institute of Technology (IRB 0481-18-RMB). This study complies with all relevant ethical regulations for studies involving human subjects. Mice brain extracts were obtained from Yaron Fuchs Laboratory, Technion, Israel.

**Cell lines**. HeLa (CCL-2), Hek293 (CRL-1573), HepG2 (HB-8065), MiaPaca2 (CRL-1420), HCT116 (CCL-247), Panc-1 (CRL-1469), A549 (CCL-185), and WM266-4 (CRL-1676) cells (purchased from ATCC, USA) were cultured in Dulbecco's modified Eagle's medium (DMEM, high glucose, Biological Industries USA, Inc., USA) supplemented with 10% (v/v) dialyzed fetal bovine serum (Hyclone Laboratories Inc., USA), 4 mM glutamine, 100 U mL$^{-1}$ penicillin, and 100 μg mL$^{-1}$ streptomycin. Jurkat (CRL-2899) and CCRF-CEM (CCL-119) cells (purchased from ATCC, USA) were cultured in RPMI medium (Biological Industries USA, Inc., USA) supplemented with 10% (v/v) dialyzed fetal bovine serum (Hyclone Laboratories Inc., USA), 4 mM glutamine, 100 U mL$^{-1}$ penicillin, and 100 μg mL$^{-1}$ streptomycin.

All cell lines were cultured using standard procedures in a 37 °C humidified incubator with 5% CO$_2$. Cell lines were tested for mycoplasma using EZ-PCR mycoplasma detection kit (Biological Industries USA, Inc., USA). We seeded 1.5 million from each cell line in six-well plates (three repeats for the cell type) with 6 mL of the media and grew them for 24 h before metabolite extraction. HCT116 were cultured using standard procedures in a 37 °C humidified incubator with 5% CO$_2$ in high-glucose DMEM supplemented with 10% heat-inactivated fetal bovine serum, 2 mM glutamine and 100 U mL$^{-1}$ penicillin, 100 μg mL$^{-1}$ streptomycin. For the hypoxic experiments, $3.5 \times 10^5$ cells in 6 cm plate were plated under normoxia conditions. Twenty-four hours after, half of the plates were moved in a humidified Whitley H35 Hypoxistation chamber (Don Whitley Scientific, Shipley, UK) at 37 °C with 1% O$_2$, 5% CO$_2$, and 94% N$_2$ for additional 38 h, another half were incubated under 21% for the same time period.

**Sample preparation**. To extract metabolites and lipids from serum samples, we mixed 20 μL of serum with an extraction solution for metabolomics analysis and 10 μL for lipidomics in 96-deep well plates. For lipidomics analysis, we utilized 100 μL of 2-propanol/methanol (6:1, v/v; slightly modified method[34]); and for metabolomics analysis, 100 μL of methanol/acetonitrile/water (5:3:1, v/v/v). After 10 min of vortexing, 800 rpm, precipitated proteins were separated by centrifugation for 20 min at 4 °C and 4000 rcf; supernatants were stored at −80 °C prior the analysis (Supplementary Fig. 13).

To extract metabolites from cancer cells, the previously published protocol[35] was employed. Metabolite extraction from media was obtained by mixing 50 μL of culture media with 200 μL of methanol/acetonitrile (5:3, v/v) solution, prechilled at −20 °C. For the measurement of intracellular metabolites, cells were washed with 2 mL of ice-cold PBS twice on ice. The cells were extracted quickly in 5 mL volume of methanol/acetonitrile/water (50:30:20, v/v/v) solution at −20 °C on dry ice by scraping. All of the metabolite samples were stored at −80 °C for at least 2 h. Protein free metabolite extractions were prepared by spinning the samples at 20,000 g for 20 min at 4 °C twice. Samples were subsequently analyzed using FI-MS.

**Flow-injection mass spectrometry**. The analysis was performed using a Q Exactive Hybrid Quadrupole Orbitrap high-resolution mass spectrometer with an ESI source (Thermo Fisher Scientific, Inc., USA). Samples were taken from an autosampler and directly injected to the mass spectrometer via an HPLC system (Ultimate 3000 Dionex LC system, Thermo Fisher Scientific, Inc., USA). All method files were written and executed via Thermo Xcalibur 4.0 software (Thermo Fisher Scientific, Inc., USA).

One stainless steel capillary ($d$—130 μm, $l$—900 mm) was used for the connection of the LC system's sample injection unit (injection volume—5 μL) with ionization source of mass spectrometer. A gradient of flow rate (Optimization of gradient see below, Supplementary Fig. 14) was employed: after a linear decrease from 0.8 to 0.075 mL min$^{-1}$ for 0.02 min constant flow rate of 0.075 mL min$^{-1}$ for 0.31 min was used for the analysis. For the washing of the system, a flow rate of 1.2 mL min$^{-1}$ was applied from 0.34 to 0.44 min followed by re-equilibration to the starting condition resulting in a total run time of 0.45 min per sample. The mobile phase consisting of 2-propanol/water (50:50, v/v) buffered with 10 mM ammonium carbonate at pH 9 for metabolomics samples and 2-propanol/water (90:10, v/v)

buffered with 10 mM ammonium carbonate at pH 9 for lipidomics samples. Mass spectra were recorded from 0.07 to 0.32 min with 3 micro scans, the AGC target value—$5 \times 10^6$ and resolution—70,000. Eight scans in negative ionization mode followed by eight scans in positive ionization mode were used for the analysis of one sample within 15 s of total scanning time (Supplementary Fig. 15). Gases flow rates in the ionization source were optimized during the preparation of mass spectrometric analysis. Due to highly different flow rates during sample examination, two different tune files were employed in order to prevent the ionization source from getting wet: 10 units of auxiliary gas and 40 units of sheath gas before injection and after 0.33 min of analysis, and with 0 units of auxiliary gas and 10 and 15 units of sheath gas for lipidomics and metabolomics samples, respectively, from injection of sample and to 0.33 min of analysis. Capillary temperature was set to 350 °C for metabolomics samples and 400 °C for lipidomics samples. Spray voltage was set to ESI (−)—3.30 kV, ESI (−)—3.80 kV and ESI (−/+)—3.75 kV for metabolomics and lipidomics analysis, respectively. The duty time of the entire method (i.e., time between consecutive injections) is ~30 s.

For all FI-MS experiments requiring more than eight scan ranges, we applied the above method consecutively while splitting the set of desired ranges to groups of eight or less scans each. Performing more than 16 scans (8 in positive and 8 in negative ionization models) with a single injection would require a flow rate lower than 75 μL min$^{-1}$, which decreases the TIC and the overall sensitivity. For example, to determine the distribution of reproducible $m/z$ features in the range from 70 to 2500 $m/z$, based on 122 scan ranges of size 20 (with 4 $m/z$ overlap between consecutive ranges), we repeatedly applied the above FI-MS method 16 times, each with a different set of 8 scan ranges.

**Optimization of FI-MS gradient of glow rate**. We aimed to derive a flow-injection method with stable TIC for 16 s a minimal total cycle time; 16 s are sufficient for eight scans in negative and eight in positive ionization models, with a resolution of 70,000, each scan with 3 micro scans (using a Q Exactive Orbitrap MS). To estimate the system dead volume (eluent volume ~65 μL) and washing volume (solvent volume ~250 μL required for washing of the system after injection), we performed measurements of metabolomics extract in isocratic elution mode with flow rate of 75 μL min$^{-1}$ (Supplementary Fig. 14a). Dead volume and washing volume typically depend on type of connectors between LC injection port and ESI source (in our case, with a single stainless steel capillary: $d$—130 μm, $l$—900 mm).

To minimize the total cycle time, we utilize a maximal possible flow rate of 1.2 mL min$^{-1}$ for washing of the system after completing MS scanning. Drastic changes in eluent flow rates require changing the flow rate of sheath and auxiliary gases (high gas flow rates is needed for evaporating eluent when its flow rate is high; low gas flows are needed for stable TIC scanning when eluent flow rate is low). Toward this end, we configure two mass spectrometer tune files, switching to low gas flow rates (0 units—auxiliary, 10 and 15 units—sheath for lipidomics and metabolomics analysis, respectively) a time 0 min, and then to high gas flow rates (10 units—auxiliary, 40 units—sheath) in 0.32 min (Supplementary Fig. 14b). After switching to low gas flow rates at 0 min, the system requires 0.08 min to stabilize flow rates; and hence we start MS scanning after 0.08 min.

The final flow rate gradients for the 0.45 min method were determined based on the dead and wash volumes and time required for gases equilibration (Supplementary Fig. 14c). Applied to analyze a serum sample, TIC remain stable within 16 s (after the initial eluent and gases equilibration 0.08 min stage; Supplementary Fig. 14d).

**Investigating ion suppression and ion competition effects**. We aimed to assess the extent of ion suppression in the ESI source versus ion competition in the Orbitrap detection system, as a mean to evaluate the potential of spectral stitching to improve the sensitivity of FI-MS analysis. Toward this end, we evaluated the ion suppression and competition effects by injecting metabolite extracts from a serum sample, while gradually inducing the ion flow by adding increasing concentrations of different compounds: 10–250 μM SDS, TC, and MRFA peptide; 10–1500 μM caffeine; 10–3000 μM β-NAD. SDS and TC were used for investigation of ion suppression and competition effects in negative ionization mode (in both FI-MS-based metabolomics and lipidomics analysis) and caffeine, MRFA and NAD in positive ionization mode (in FI-MS-based metabolomics analysis; Supplementary Figs. 1 and 2).

For each of the above compounds, we performed a series of FI-MS runs while gradually increasing its concentration in the analyte (considering 8–11 concentrations per compound within the ranges specified above; with six repeated injections of the analyte for each concentration), configuring a 20 $m/z$ scan range which excludes this compound and an overlapping 24 $m/z$ scan range that includes it. We repeated the experiment twice: once, with both the 20 $m/z$ and 24 $m/z$ scan ranges starting at the $m/z$ of the ion of added compound minus 22 $m/z$ units; and once, when the 20 $m/z$ and 24 $m/z$ ranges end at the $m/z$ of the ion of added compound $m/z$ plus 22 $m/z$ units (see Supplementary Fig. 3). The effect of ion suppression in the ESI was assessed based on the drop in the number of reproducibly detected $m/z$ features in the 20 $m/z$ scan range (which excludes the $m/z$ signal of the added compound) as higher and higher concentrations of the compound were added to the analyte; for every concentration of the added compound, the set of reproducibly detected $m/z$ features was determined based on

the six repeated injections as defined below. The total effect of ion competition in the detection system and ion suppression in ionization source was assessed based on the drop in the number of detected $m/z$ features when configuring a 24 $m/z$ range (that includes the $m/z$ signal of the added compound; considering only $m/z$ features within the smaller 20 $m/z$ interval without the added compound).

**Liquid chromatography mass spectrometry.** Chromatographic separation for polar metabolites was achieved on a SeQuant ZIC-pHILIC column (2.1 × 150 mm, 5 μm, EMD Millipore) for polar metabolites. Flow rate was set to 0.2 mL min$^{-1}$, column compartment was set to 30 °C, and autosampler tray maintained 4 °C. Mobile phase A consisted of 20 mM ammonium carbonate and 0.01% (v/v) ammonium hydroxide. Mobile Phase B was 100% acetonitrile. The mobile phase linear gradient (%B) was as follows: 0 min 80%, 15.0 min 20%, 15.1 min 80%, 23.0 min 80%. A mobile phase was introduced to Thermo Q Exactive mass spectrometer with an ESI source working in polarity switching mode. Ionization source parameters were following: sheath gas 25 units, auxiliary gas 3 units, spray voltage 3.3 and 3.8 kV in negative and positive ionization mode, respectively, capillary temperature 325 °C, S-lens RF level 65, auxiliary gas temperature 200 °C. Metabolites were analyzed in the range 72–1080 $m/z$.

For nonpolar metabolites, a Kinetex reversed phase C18 column (3 × 150 mm, 2.6 μm, Phenomenex) was used. Flow rate was set to 0.5 mL min$^{-1}$, column compartment temperature controlled at 65 °C, and autosampler tray maintained 10 °C. Mobile phase A consisted of 10 mM ammonium formate in acetonitrile/water (60:40, v/v) solution; mobile phase B consisted of 10 mM ammonium formate in 2-propanol/acetonitrile/water (90:8:2, v/v/v) solution. For separation, the following multi-step gradient (%B) was used: 0 min 15%, 2.0 min 30%, 2.5 min 48%, 11.0 min 82%, 11.5 min 99%, 20 min 99%, 20.1 min 15%. Total analysis time 25 min. Eluent was analyzed by Thermo Q Exactive mass spectrometer with an ESI source working in polarity switching mode. Ionization source parameters were following: sheath gas 50 units, auxiliary gas 15 units, spray voltage 3.5, and 3.3 kV in negative and positive ionization mode, respectively, capillary temperature 350 °C, S-lens RF level 70, auxiliary gas temperature 350 °C. Metabolites were analyzed in the range 150–2000 $m/z$. Metabolite retention times were determined by analyzing pure chemical standards, for analysis of LC-MS data MAVEN[29] 6.2 software was used.

**Data processing and analysis of reproducible $m/z$ features.** To determine the number of reproducibly detected $m/z$ features by a specific FI-MS method configuration, we performed six repeated injections of the biological sample from the same vial followed by the injection of six blank samples (i.e., sample preparation protocol applied to a water sample) and analyzed the measured data as following: we converted raw mass spectrometer measurement files to mzXML using ProteoWizard[36]. Then mzXML files were loaded and further analyzed using Matlab 2017b (The MathWorks, Inc., USA); all code was made publicly available as a GitHub repository at https://github.com/shovall/FlowInjectionMSOptimization. $m/z$ signals of all 12 injections were grouped based on a tolerance of 5 ppm. Then, reproducible $m/z$ features were identified based on the following criteria (in accordance with previous studies[14,37,38]): (1) observed in 90% of the biological sample injections; (2) the median intensity of the biological sample injections is above 1000 units; (3) signal-to-noise ratio above 4; i.e., the median intensity in the biological sample injections divided by the maximal intensity of the blank injections. (4) RSD across the biological sample injections lower than 30%. The annotation of the reproducible $m/z$ features was done using publicly available databases, LMSD[26] for lipidomics analysis, and HMDB[25] for metabolomics analysis. The annotation process was based on accurate mass considering $[M + H]^+$ and $[M - H]^-$ adducts, accounting for a tolerance of 5 ppm. The mass accuracy was measured based on injection of internal standards (Supplementary Table 5).

**A method for finding optimized scan ranges.** Given an $m/z$ interval of interest, denoted $[\alpha, \beta]$, we run FI-MS analysis while configuring the quadrupole to separately pass ions within a series of consecutive 20 $m/z$ scan ranges, altogether covering the entire scan range (i.e., $\frac{\beta - \alpha}{20}$ scan ranges; performing six repeated measurements of each scan range). The reproducible features in each 20 $m/z$ scan range are determined as described above. We denote by $X_i$ the number of reproducible features found in the scan range $[i, i + 1]$, and by $C_i$ the cumulative number of reproducible features in the scan range $[\alpha, 1]$, computed by $C_i = \sum_{j=\alpha}^{i-1} X_j$. We aim to determine the boundaries of $n$ scan ranges with an equal number of reproducible $m/z$ features, denoting the $j$'th range by $[r_{j-1}, r_j]$, where $r_0 = \alpha$ and $r_n = \beta$. For every $j \geq 1$ and $j < n$, we set: $r_j = \min(\{i \mid C_i > j \frac{C_\beta}{n}\})$.

**Amino acids and lipids measurements in serum samples.** We applied LC-MS and FI-MS to quantify the absolute concentration of nine amino acids (alanine, asparagine, aspartic acid, valine, lysine, phenylalanine, proline, tryptophan, and tyrosine) in Human AB Serum (Biological Industries USA, Inc., USA). We analyzed metabolite extracts with increasing concentrations of internal standards for all amino acids (see Supplementary Table 2a) using LC-MS and FI-MS, calculating absolute concentrations via the standard addition method; using linear regression to fit the measured intensities with the added metabolite concentrations and

extrapolating to the point of zero intensity (calculating a 95% confidence intervals). We utilized the LC-MS method with chromatographic separation for polar metabolites, as described above; and FI-MS-based metabolomics.

We applied LC-MS and FI-MS to quantify the concentration of two lipids (18:1 lyso phosphocholine and 18:1 lyso phosphoethanolamine) in the same commercial serum sample. We analyzed metabolite extracts with increasing concentrations of stable isotopic standards of these compounds with LC-MS and FI-MS and quantified the absolute concentrations via internal calibration curves (Supplementary Table 2b). We utilized the LC-MS method with chromatographic separation for nonpolar metabolites, as described above; and FI-MS-based lipidomics.

**FI-MS analysis of 98 serum samples of healthy individuals.** We analyzed 98 serum samples from healthy individuals using our optimized ranges FI-MS method metabolomics method, uniform ranges FI-MS, and single range FI-MS. Samples were obtained from the Israeli Midgam Biobank (IRB: 0481-18-RMB). Sample preparation for metabolomics analysis was performed as described above; adding a mix of internal standards to the extraction solution, enabling to infer MS mass accuracy (μM of $^{13}C_{11}$-tryptophan (MW—215 Da), 0.5 μM of puromycin dihydrochloride (MW—471 Da), and kiton red 620 (MW—580 Da)); inferred mass accuracy was used for metabolite annotation. Quality control (QC) samples were prepared by mixing 20 μL aliquots from each sample. A QC sample was injected every 5th serum sample and a blank sample every 10th serum sample. Every sample was injected four times, resulting in a total of 600 injections for each of the metabolomics methods (including blank and QC sample injections).

**Gender prediction via data obtained with FI-MS methods.** We aimed to reproduce an analysis of gender prediction performed with LC-MS-based metabolomics analysis of serum samples[28], using our and current FI-MS methods. Reproducibly detected $m/z$ features were identified based on repeated analysis of the QC samples. Gender prediction was performed using a random forest model, consisting of 100 decision trees[39] constructed using the identified significant $m/z$ features. Model accuracy was calculated based on out-of-bag observations.

**Clustering ions associated with distinct metabolites.** For measurements performed with each of the three FI-MS methods (our optimized ranges FI-MS, uniform ranges FI-MS, and single range FI-MS), we clustered the set of identified $m/z$ features based on a high correlation in the measured intensity profiles across the 98 analyzed samples. This was done using a greedy approach: starting with a set of reproducibly detected $m/z$ features by the FI-MS method (determined based on repeated FI-MS analyses of QC samples) and an empty set of clusters, and then iterating over the set of $m/z$ features and adding them to either an existing or a new cluster. An $m/z$ feature is added to an existing cluster in case the Pearson correlation between its intensity profile across samples and the intensity profile of an $m/z$ feature in the cluster is above a predefined threshold; alternatively, the $m/z$ feature is added to a new cluster.

Considering a threshold of a minimal Pearson correlation of 0.95, the derived $m/z$ feature clusters were indeed enriched with different isotopic forms and adducts of the same metabolite; in both positive and negative ionization modes, the most frequent difference in $m/z$ values between features in a cluster is 1.003 $m/z$ due to the natural abundance of 13C and; and in positive ionization mode, a frequent difference in $m/z$ values between features in the same cluster is 21.985 $m/z$ for adduct $[M + Na]^+$; and 18.010 $m/z$ in negative mode for adduct $[M - H_2O - H]^-$.

**DDA FI-MS analysis of serum samples.** We performed FI-MS analysis with DDA MS/MS mode on Human AB Serum (Biological Industries USA, Inc., USA), splitting the $m/z$ interval from 70 to 990 $m/z$ to 13 ranges of 80 $m/z$ width. For each range, we performed a separate FI-MS analysis, once in negative and once in positive ionization modes (using the same injection volume, flow gradients, and MS ionization source parameters as described in the paper for our optimized ranges FI-MS method). Carrying out a longer injection with ESI by slowing down the flow rate would lower the sensitivity of the method and damage reproducibility.

$MS^2$ data were obtained for a total of 36 and 47 precursor ions in negative and positive ionization modes respectively (with at least a single ion in the $MS^2$ spectra with an intensity > 10,000). Annotation of precursor ions was performed based on a match with publicly available METLIN $MS^2$ spectra (using the online fragment similarity search tool) for polar compounds, and LMSD bulk Structure Search and LMSD Product ion calculation tool for prediction of MS/MS fragments for nonpolar compounds. Utilizing METLIN $MS^2$ spectra, we annotated 22 polar precursor ions, 20 with a unique hit (with a match in 1–4 collisional fragments) and 2 with 2 possible hits due to isomers. Using LMSD, we annotated another 61 nonpolar compounds (Supplementary data 5).

**Reporting summary.** Further information on research design is available in the Nature Research Reporting Summary linked to this article.

## Data availability

The metabolomics data generated and analyzed in this study is available in Metabolomics Workbench with the identifier ST001380 [https://doi.org/10.21228/M8P41V]. The source data underlying Figs. 1b–i, 2b–f, 3, and 4 and Supplementary Figs. 1, 2, 5–12, and 14a, b, d are provided as a Source Data file. All other data are available from the corresponding author on reasonable request. The Human Metabolome Database (HMDB)[25] and LIPID MAPS Structure Database (LMSD)[26] were obtained from https://hmdb.ca/ and https://www.lipidmaps.org/, respectively. Source data are provided with this paper.

## Code availability

All code is publicly available as a GitHub repository at https://github.com/shovall/FlowInjectionMSOptimization Source data are provided with this paper.

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

## Acknowledgements

We thank Dr. Markus Himmelsbach, Dr. Dmitry Eremin, Dr. Andrey Stavrianidi, Ivan Plyushchenko, and Ksenia Kogan for helpful feedback and discussions; Prof. Yaron Fuchs and Egor Sedov for providing mice tissue extracts. The research leading to these results has received funding from the European Research Council/ERC Grant Agreement No. 714738, and from the ISRAEL SCIENCE FOUNDATION (grant No. 3442/19), within the Israel Precision Medicine Partnership program.

## Author contributions

B.S. and S.L. contributed equally to this work. B.S., S.L., and T.S. designed and planned the study, analyzed data, and interpreted results. P.K. and E.A. performed tissue culture experiments. B.S. performed sample preparation and FI-MS analysis. S.L. and T.S. performed the computational analysis. B.S. and N.S. performed DDA FI-MS/MS analysis and metabolite annotation. LC-MS system was configured by D.M. All authors contributed to the writing of the paper.

## Competing interests

The authors declare the following competing interests: B.S., S.L., and T.S. declare that the results presented in this work (FI-MS scan ranges optimization) were included in a submitted Provisional Patent application (#62911629). The other authors declare no competing interests.
