## [Peer Review File · Nature Communications]

Reviewers' comments:

Reviewer #1 (Remarks to the Author):

The authors of this manuscript present a new approach to overcome a major limitation in current use of Orbitrap technology in non-targeted molecular profiling. The contribution and presentation is rather technical, but I believe their approach is smart and this work has the potential to advance metabolomics/lipidomics profiling with Orbitrap mass analyzers in large sample cohorts as often necessary to answer biological or clinical questions.

I compliment the authors for a thorough effort to pinpoint the actual bottleneck of Orbitrap technology in high-throughput spectral profiling. A similarly systematic analysis was missing in the field. Overall, the manuscript is well-written and easy to follow for a more expert reader. However, to make the significance of their analyses evident for readers/users with a broader expertise, I would recommend a clarification and/or more explicit explanation in several places, as also indicated below.

Several points should be addressed before publication of this manuscript:

(1) Conceptually, the approach that the authors take is valid, and they come up with an appropriate solution for the ion competition problem. What is missing is a better argument for readers to judge the real value of optimizing and using the authors' measurement routine in metabolome/lipidome analyses. For example, can the authors provide any kind of conclusion of serum/cancer cell measurements? Perhaps expected or previously found differences/trends that the authors were able to see also in their analysis, even just on a very descriptive level. It is one thing to measure more features, but the other question is always whether the features measured are also informative when it comes to biological interpretation.

(2) While the authors indeed contribute an original improvement specifically for Orbitrap analyzers, other analytical platforms are already capable of FI-MS-based metabolomics, as correctly referenced by the authors in the introduction. This should also be reflected in the final conclusion, i.e. lines 249-253 should mention Orbitraps, and please provide a more precise discussion of the specific novelty and benchmarks.

(3) In the first part of the Results section, it would arguably be just as interesting to explicitly analyze how the intensity distribution changes, i.e. whether low-abundant or poorly ionized metabolites are

a priori more at risk to be lost? What I would perhaps expect is that the mean or median intensity of detected ions increases when the larger scan range is chosen, as a result of low-abundant ions not being sampled at sufficiently high levels during the short injection time of the c-trap.

(4) Similarly, could the authors perhaps comment (e.g. in the conclusion) to what extent their optimization will generalize, e.g. whether/how they expect the scan ranges to generalize to other sample types. How sensitive are the 8 scan ranges they found for serum samples to changes in the molecular composition of a sample, or when analyzing a new sample type? Should scan ranges be optimized for each new sample type or even sample group?

(5) Given that the authors specifically chose serum samples for method development and as a proof of concept, their analysis should also have included the NIST standard reference material SRM-1950 (Metabolites in human plasma) as a benchmark for analytical performance in a complex matrix.

(6) For sake of reproducibility the authors should add more information on how the initial exhaustive scanning is implemented. From the methods section, it is not clear how the authors achieved the required “run times in the order of minutes” (line 151)? This is typically not feasible with standard flow-injection approaches, and as far as I can tell also not with the injection volume and flow rate scheme described in the methods section. The sample would pass the ion source in a much shorter time frame, and there is no mention of e.g. repeated injections.

(7) Related to the previous point, the measurement routine described in the methods section seems overly complicated (varying flow rates and ESI parameters). Could the authors perhaps comment on whether these conditions are crucial, how long the transient signal is seen (what does the TIC of one injection look like?) and how the performance would change if a constant flow rate is employed (avoiding changing ionization conditions)?

(8) Clear conclusions should be added or simplified in some sections to accommodate readers with a broader expertise. E.g. analytical performance is assessed and the analytical figures of merit are described technically, but not in terms of consequences/benefits of a low RSD (line 215) and high linear dynamic range (line 222).

In lines 168-169, the description of different challenges in high- vs. low-density spectral regions reads rather convoluted and could be replaced by a more general/conceptual statement. E.g. low-density spectral regions tolerate wider scan ranges (i.e. shorter measurement times), but in mass ranges with a high ion density dense sampling is crucial.

(9) Please mention somewhere explicitly in what aspects the metabolomics and lipidomics methods differ. The authors distinguish between the two (line 140) already before defining optimal scan ranges, but it is not so clear what makes each method a metabolomics or lipidomics method (only ESI parameters?).

(10) Terminology/wording:

a. The authors should refer to their approach as “flow injection”, and not as “direct injection” (e.g. in the abstract) to avoid confusing flow injection with direct infusion.

b. The phrase in line 176 “...that are expected to maximize the total number of significantly detected m/z features” is vague, a more explicit description like in line 181-182 would be a better introduction.

c. The way the authors refer to the number of scan ranges in lines 182 and following is potentially misleading and should be revised, e.g. “8 m/z scan ranges” could be understood as scan windows with a width of 8 m/z, rather than 8 scan windows of optimized width. Similarly in line 200 “We found that 4 m/z scan ranges are sufficient ...”, etc.

Reviewer #2 (Remarks to the Author):

This paper describes an extension of an existing method (direct infusion mass spectrometry spectral stitching metabolomics / lipidomics) where the authors have optimised one parameter to enhance detection sensitivity. It is shown here that if the 8 detected m/z windows are adjusted in width to each detect the same number of features, then the detection sensitivity is significantly enhanced. This method also allows the detection time to be shortened, which allows higher throughput. This method will be useful for researchers in the field of direct infusion MS metabolomics and lipidomics. A key result in the paper is the demonstration that ion competition in the detection system has a much greater negative impact on sensitivity compared to ion suppression. This finding is relevant and interesting to both to direct infusion MS users and also LC-MS metabolomics and lipidomics

users. Overall the written text and figures are clear, and the data shown mostly supports the findings.

Specific points:

General: Only two compounds were chosen to demonstrate the effects of ion suppression: SDS and taurocholic acid. More compounds should be tested to prove that this phenomenon is not compound specific. Do you have any evidence to show that ion suppression is only caused by ease of compound ionisation? i.e. would you expect all compounds that ionise easily to have similar levels of ion suppression? Would other matrix effects such as salt content alter ion suppression? Did you find that flow rate influences ion suppression (the infusion flow rate here is relatively high. Nanoflow has been shown to reduce ion suppression relative to $\mu\text{L}/\text{min}$ flow)?

General: It is unclear what the final pipeline is for data acquisition in a biological study. Please define this. E.g. when using the final method for biological studies, do you acquire each sample multiple times or just once? For the test data, samples were acquired 6 times in order to filter and retain high quality peaks. In practice when using this method, is there a need for technical replication of each biological sample to determine quality peaks from bad peaks. If so how does this increase the acquisition time for each sample? Can you clearly define the time length for analysing one biological sample (include any technical replicates and also instrument time between samples).

Abstract, L17 – You state that LC-MS is not suitable for high throughput applications. This depends on how high-throughput is defined. 15 min LCMS metabolomics assays are fairly common and this can be used to analyse 1000+ samples which is normally considered high throughput. Also it is a trade-off, LCMS will give more structural information on compounds and separate isobaric compounds. Can you define what you consider high throughput and also discuss the trade-off between structural information and time (LCMS v FIMS).

Abstract – insufficient method information is currently in the abstract, especially as this is a methods paper.

Abstract, L26 – method is said to be 15 s long, is this correct? I'm not clear if the widows are 15s or the overall method. Also how long is instrument time between samples?

Abstract, L26 –the term quantification could be misleading as metabolomics is considered semi-quantitative rather than fully quantitative.

L37 – metabolomics should only be described as measuring ‘concentration’ if a chemical standard, internal standard and standard curve is also used.

L43 – I wouldn’t describe NMR as high resolution. Also for MS high resolution only applies to some detectors.

L48 – LCMS metabolomics assays for high throughput typically take 15 min with LC standard columns. These can be used for high throughput (depending on how high throughput is defined, see first point above).

L68-70 A method taking a few minutes would normally still be considered high throughput.

L83 is your method 2 mins in length, i.e. 15 s x 8, or 15 s as suggested by the abstract?

L114 Section 2.5 doesn’t exist in the paper. Should this be ‘the methods section’?

L115-119 When considering detection sensitivity between 20 m/z and 24 m/z windows, did you only compare the features common to both windows (i.e. only those that fell within the 20 m/z windows)? This would be the fairest comparison.

L162-174 When counting features in the increasing window sizes, did you count the features in the entire window, or within a region common to all windows? The latter would be the fairest comparison.

L187-190 Again I’m not fully clear on total acquisition time for each method.

L226 & L245 Define annotated. Were these putatively annotated based solely on accurate mass or was MS/MS also used. If it was accurate mass only, then they should be defined as putatively annotated.

L227-229 Was any extra annotation work done on these peaks other than accurate mass matching? Without this (e.g. MSMS annotations) you cannot be sure these actually are fatty acids, steroids and carbohydrates.

L300 Did you check for cross contamination between samples? A good way to test would be running 2-3 blanks after a sample to see the degree of carry-over.

L307-311 Why was there a gradient of flow? When you acquire data is the flow rate constant?

L318 On the Q Exactive HESI source 3 gasses exist: sheath, aux, sweep. Is your method applicable for use on the HESI source? Which of these are the drying and nebuliser gasses?

L342: 5ppm is a large mass tolerance when annotating based on only accurate mass. This will lead to many false positive identifications. The Q Exactive generally performs better than +/- 5ppm, have you assessed your system and considered decreasing the ppm tolerance window?

Figure 1 b,c & Fig 3 b, c: Add the name of the metabolite to the x axis to make reading easier.

Points related to data processing and raw data:

L329: mzXML is a relative old data format / standard and is not maintained and further developed anymore. To make the implementation future proof the workflow should be compatible with mzML (more common format in Metabolomics and Proteomics) and mzXML formatted files.

L332: The GitHub repository is not available via the link provided and therefore the implementation cannot be tested. Additionally, Matlab is not open source and a licence is required to test the code / implementation. This is a significant limitation and will limit the usability of the code/workflow provided. Therefore, consider other already existing tools written in open-source languages, such as R and python.

L336: Define how a signal-to-noise ratio threshold of 4 was defined. What optimisation was used to define this threshold?

It is unclear from the manuscript if the raw data has been submitted to one of the main metabolomics repositories, including Metabolights. I highly recommend making the raw data publicly available.

We would like to thank the reviewers for commenting on our manuscript. The manuscript has been revised based on these comments, as described below.

Reviewer #1 (Remarks to the Author):

The authors of this manuscript present a new approach to overcome a major limitation in current use of Orbitrap technology in non-targeted molecular profiling. The contribution and presentation is rather technical, but I believe their approach is smart and this work has the potential to advance metabolomics/lipidomics profiling with Orbitrap mass analyzers in large sample cohorts as often necessary to answer biological or clinical questions.

I compliment the authors for a thorough effort to pinpoint the actual bottleneck of Orbitrap technology in high-throughput spectral profiling. A similarly systematic analysis was missing in the field. Overall, the manuscript is well-written and easy to follow for a more expert reader. However, to make the significance of their analyses evident for readers/users with a broader expertise, I would recommend a clarification and/or more explicit explanation in several places, as also indicated below.

We would like to thank the reviewer for the kind compliments on our manuscript. In order to accommodate readers with a broader expertise, several such clarifications were added (see pages 7 and 9; and replies below). The Methods Section is now further elaborated.

Several points should be addressed before publication of this manuscript:

(1) Conceptually, the approach that the authors take is valid, and they come up with an appropriate solution for the ion competition problem. What is missing is a better argument for readers to judge the real value of optimizing and using the authors' measurement routine in metabolome/lipidome analyses. For example, can the authors provide any kind of conclusion of serum/cancer cell measurements? Perhaps expected or previously found differences/trends that the authors were able to see also in their analysis, even just on a very descriptive level. It is one thing to measure more features, but the other question is always whether the features measured are also informative when it comes to biological interpretation.

Following the reviewer's comment, we applied our FI-MS method to reproduce results from two recent studies involving metabolomics analysis of serum and cancer cells:

We applied our FI-MS method to analyze inter-subject variability in serum metabolome within a group of 98 healthy individuals (Supplementary Methods). Our analysis shows variability in the abundance of ~3500 m/z features across the analyzed population (i.e. number of m/z features whose inter-subject RSD is 50% higher than the RSD in repeated injection of QC samples, Dunn et al., 2014); this number of biologically important m/z features is only ~20% lower than that detected in a recent study via standard LC-MS analysis, though the FI-MS analysis is ~100-fold faster (Dunn et al., 2014). We find that the distribution of inter-subject RSD is skewed to lower values, in accordance with the LC-MS based results (Supplementary Figure 7, Supplementary Data File 2, Dunn et al., 2014); with a similar median inter-subject RSD of ~40% and maximum RSD of ~700%. Specifically, in accordance with the LC-MS results, we find that aromatic amino acids (tryptophan, phenylalanine and tyrosine) have low inter-subject RSD (< ~35%); fatty acids (tetradecanoic and hexadecenoic acids) have intermediate RSD values (66% and 71%, respectively); and, expectidely, drug metabolites (paracetamol, sitaxentan) have the highest RSD (> 300%). Utilizing the entire set of serum FI-MS measurements to reproduce a study of gender specific metabolite fingerprints performed with LC-MS, resulted in overall similar results (Supplementary Figure 8).

Supplementary Figure 7 Distribution of RSD values for significant m/z features measured across 98 serum samples of healthy individuals in negative (a) and positive (b) ionization modes.

As a model system for testing whether our FI-MS method could capture important biological alterations in cellular metabolome, we reproduced a study on the effect of oxygen level on intracellular metabolites performed via LC-MS (Frezza et al., 2011). We applied the developed FI-MS method to measure the metabolome of HCT116 cancer cells grown in hypoxia (1% O_2 , see Methods) and normoxia (20% O_2). For a set of 31 metabolites whose concentration was previously reported to increase under hypoxia based on LC-MS, our FI-MS analysis shows significantly higher fold-changes in abundance (in hypoxia versus normoxia) compared to other detected m/z features (Wilcoxon rank sum test, p -value < 10^{-7}).

(2) While the authors indeed contribute an original improvement specifically for Orbitrap analyzers, other analytical platforms are already capable of FI-MS-based metabolomics, as correctly referenced by the authors in the introduction. This should also be reflected in the final conclusion, i.e. lines 249-253 should mention Orbitraps, and please provide a more precise discussion of the specific novelty and benchmarks.

We now explicitly state in the Discussion that our method optimization is strictly for Orbitrap MS.

(3) In the first part of the Results section, it would arguably be just as interesting to explicitly analyze how the intensity distribution changes, i.e. whether low-abundant or poorly ionized metabolites are a priori more at risk to be lost? What I would perhaps expect is that the mean or median intensity of detected ions increases when the larger scan range is chosen, as a result of low-abundant ions not being sampled at sufficiently high levels during the short injection time of the c-trap.

We thank the reviewer for this suggestion. Expectedly, the ion competition effect observed when scanning for the 24 m/z range spanning taurocholic acid masks the detection of low-abundant and/or poorly ionized compounds having low intensity (Figure 1d-f); gradually increasing the concentration of taurocholic acid leads to loss of m/z features having higher and higher intensities (Figure 1d).

Figure 1 d-f Ion competition in the detection system rather than ion suppression in ESI is the prime reason for the reduced sensitivity of FI-MS based metabolomics and lipidomics. **d** The median intensity of m/z features (measured without adding TC; y-axis) who become undetected when adding increasing concentrations of TC (x-axis), scanning for the 20 m/z scan range (in blue) and for the 24 m/z scan range (in red); black horizontal line represents the median intensity of m/z features detected without adding TC. **e, f** Ion intensity (y-axis) and m/z (x-axis) detected in serum when scanning for the 20 m/z scan range (**e**) and the 24 m/z scan range (**f**); ions undetected when adding 100 μM of TC are marked with red crosses.

(4) Similarly, could the authors perhaps comment (e.g. in the conclusion) to what extent their optimization will generalize, e.g. whether/how they expect the scan ranges to generalize to other sample types. How sensitive are the 8 scan ranges they found for serum samples to changes in the molecular composition of a sample, or when analyzing a new sample type? Should scan ranges be optimized for each new sample type or even sample group?

Following the reviewer's comment, we now applied the scan range optimization method to several serum samples from different donors; and to several different mice brain lipidomic extracts. We show that different samples of the same type produce a similar characteristic distribution of m/z features (Supplementary Figure 5); and hence scan range optimization should be performed once for every sample type of interest.

Supplementary Figure 5 Reproducibility of the distribution of significant m/z features for different samples of the same type. **a** Distribution of significant m/z features based on 64 ranges exhaustive spectral-stitching experiment (see Methods) for 6 different serum samples of healthy individuals (in blue) and 6 extracts of brain tissue of mice (in red). **b** Sets of optimized ranges for FI-MS with 8 ranges in negative ionization mode based on the obtained significant m/z features distributions.

(5) Given that the authors specifically chose serum samples for method development and as a proof of concept, their analysis should also have included the NIST standard reference material SRM-1950 (Metabolites in human plasma) as a benchmark for analytical performance in a complex matrix.

Following the reviewer's comment, we applied LC-MS to quantify the concentration of several polar and non-polar metabolites in serum samples (via two different quantification methods: standard addition and internal standards with internal isotopic compounds) and utilized that as a benchmark for the analytical performance of our FI-MS approach. All information on metabolite quantification via FI-MS and LC-MS are given in Supplementary Methods. Overall, the comparison of metabolite and lipid

concentrations measured by FI-MS to LC-MS based measurements is conceptually similar to a validation via NIST material.

Figure 3d

Correlation between FI-MS and LC-MS based concentration measurements of 9 amino acids and 2 lipids in serum.

(6) For sake of reproducibility the authors should add more information on how the initial exhaustive scanning is implemented. From the methods section, it is not clear how the authors achieved the required “run times in the order of minutes” (line 151)? This is typically not feasible with standard flow-injection approaches, and as far as I can tell also not with the injection volume and flow rate scheme described in the methods section. The sample would pass the ion source in a much shorter time frame, and there is no mention of e.g. repeated injections.

For all FI-MS experiments requiring more than 8 scan ranges, we applied our method consecutively while splitting the set of desired ranges to groups of 8 or less scans each. Performing more than 16 scans (8 in positive and 8 in negative ionization models) with a single injection would require a flow rate lower than 75 $\mu\text{L}/\text{min}$, which decreases the TIC and the overall sensitivity. For example, to determine the distribution of significant m/z features in the range from 70 to 2500 m/z , based on 122 scan ranges of size 20 (with 4 m/z overlap between consecutive ranges), we repeatedly applied the above FI-MS method 16 times, each with a different set of 8 scan ranges. This is now explained in the main text and Methods Section.

(7) Related to the previous point, the measurement routine described in the methods section seems overly complicated (varying flow rates and ESI parameters). Could the authors perhaps comment on whether these conditions are crucial, how long the transient signal is seen (what does the TIC of one injection look like?) and how the performance would change if a constant flow rate is employed (avoiding changing ionization conditions)?

We aimed to derive a flow injection method with stable TIC for 16 seconds a minimal total cycle time; 16 seconds are sufficient for 8 scan in negative and 8 in positive ionization models, with a resolution of 70,000, each scan with 3 micro scans (using a Q Exactive Orbitrap MS). To estimate the system dead volume (eluent volume ~65 μL) and washing volume (solvent volume ~250 μL required for washing of the system after injection), we performed measurements of metabolomics extract in isocratic elution mode with flow rate of 75 $\mu\text{L min}^{-1}$ (Supplementary Figure 10a). Dead volume and washing volume typically depend on type of connectors between LC injection port and ESI source (in our case, with a single stainless steel capillary: $d = 130 \mu\text{m}$, $l = 900 \text{ mm}$).

To minimize the total cycle time, we utilize a maximal possible flow rate of 1.5 mL min^{-1} for washing of the system after completing MS scanning. Drastic changes in eluent flow rates require changing the flow rate of sheath and auxiliary gases (high gas flow rates is needed for evaporating eluent when its flow rate is high; low gas flows are needed for stable TIC scanning when eluent flow rate is low). Towards this end, we configure two mass-spectrometer tune files, switching to low gas flow rates (0 units – auxiliary, 10 and 15 units – sheath for lipidomics and metabolomics analysis respectively) a time 0 minutes, and then to high gas flow rates (10 units – auxiliary, 40 units – sheath) in 0.32 min (Supplementary Figure 10b). After switching to low gas flow rates at 0 min, the system requires 0.08 min to stabilize flow rates; and hence we start MS scanning after 0.08 min.

The final flow rate gradients for the 0.45 min method were determined based on the dead and wash volumes and time required for gases equilibration (Supplementary Figure 10c). Applied to analyse a serum sample, the TIC remain stable within 16 seconds (after the initial eluent and gases equilibration 0.08 min stage; Supplementary Figure 10d).

Supplementary Figure 10 Optimization of flow rate gradient for FI-MS analysis. **a** TIC chromatogram of FI-MS serum sample analysis in isocratic mode; high and stable TIC is obtained within a 0.3 min interval (in green). **b** Sheath (in blue) and aux (in red) gases flow dynamics achieved by switching between two mass spectrometer tune files; 1st on time zero, switching to low gases flow rates that stabilizes before the beginning of scanning (where eluent flow is low; in green), and 2nd, on time 0.32 min, switching to high gases flow rates for humidity control during a washing step (where eluent flow is high). **c** An optimal gradient of eluent flow minimizing the total injection cycle time (with 75 μ L/min flow rate during scanning; in green). **d** TIC chromatogram of FI-MS serum sample analysis with the optimal gradient elution flow; high and stable TIC is obtained within a 0.25 min interval (in green).

(8) *Clear conclusions should be added or simplified in some sections to accommodate readers with a broader expertise. E.g. analytical performance is assessed and the analytical figures of merit are described technically, but not in terms of consequences/benefits of a low RSD (line 215) and high linear dynamic range (line 222).*

This is now clarified to accommodate readers with a broader expertise (lines 213-214 and 221-222).

In lines 168-169, the description of different challenges in high- vs. low-density spectral regions reads rather convoluted and could be replaced by a more general/conceptual statement. E.g. low-density spectral regions tolerate wider scan ranges (i.e. shorter measurement times), but in mass ranges with a high ion density dense sampling is crucial.

The text was modified to relate to low/high density spectral regions, as suggested.

(9) Please mention somewhere explicitly in what aspects the metabolomics and lipidomics methods differ. The authors distinguish between the two (line 140) already before defining optimal scan ranges, but it is not so clear what makes each method a metabolomics or lipidomics method (only ESI parameters?).

The metabolomics and lipidomics methods differ with respect to extraction protocol, eluent composition, and ESI parameters (as now further elaborated upon in Methods). We now refer readers to the Methods section when first mentioning the implementation of the lipidomics method. We also added a supplementary describing the sample preparation for metabolomics and lipidomics (Supplementary Figure 9).

Supplementary Figure 9 Extraction protocols of serum samples for metabolomics and lipidomics FI-MS analysis.

(10) Terminology/wording:

a. The authors should refer to their approach as “flow injection”, and not as “direct injection” (e.g. in the abstract) to avoid confusing flow injection with direct infusion.

We now specify “flow injection” in all places.

b. The phrase in line 176 “...that are expected to maximize the total number of significantly detected m/z features” is vague, a more explicit description like in line 181-182 would be a better introduction.

This is now clarified (lines 169-171).

c. The way the authors refer to the number of scan ranges in lines 182 and following is potentially misleading and should be revised, e.g. “8 m/z scan ranges” could be understood as scan windows with a width of 8 m/z, rather than 8 scan windows of optimized width. Similarly, in line 200 “We found that 4 m/z scan ranges are sufficient ...”, etc.

This was revised.

Reviewer #2 (Remarks to the Author):

This paper describes an extension of an existing method (direct infusion mass spectrometry spectral stitching metabolomics / lipidomics) where the authors have optimised one parameter to enhance detection sensitivity. It is shown here that if the 8 detected m/z windows are adjusted in width to each detect the same number of features, then the detection sensitivity is significantly enhanced. This method also allows the detection time to be shortened, which allows higher throughput. This method will be useful for researchers in the field of direct infusion MS metabolomics and lipidomics. A key result in the paper is the demonstration that ion competition in the detection system has a much greater negative impact on sensitivity compared to ion suppression. This finding is relevant and interesting to both to direct infusion MS users and also LC-MS metabolomics and lipidomics users. Overall the written text and figures are clear, and the data shown mostly supports the findings.

Specific points:

General: Only two compounds were chosen to demonstrate the effects of ion suppression: SDS and taurocholic acid. More compounds should be tested to prove that this phenomenon is not compound specific. Do you have any evidence to show that ion suppression is only caused by ease of compound ionisation? i.e. would you expect all compounds that ionise easily to have similar levels of ion suppression? Would other matrix effects such as salt content alter ion suppression?

Following the reviewer's comment, we now tested the effect of three additional compounds ion suppression and ion competition in the MS detection system (Supplementary Figure 1a). The ionization efficiency of the overall set of 5 tested compounds (together with the 2 already tested in the previous version) vary substantially (more than 1.5 orders in magnitude). Consistent with our previous observation, ion competition in the MS detection system is the major effect that limits the number of significantly detected m/z features (Supplementary Figure 1 f-k).

Supplementary Figure 1 Ion competition and ion suppression effects in the metabolomics analysis induced by adding a series of increasing concentration of two compounds, evaluated via different scan ranges. **a** The distribution of \log_{10} intensities of significant m/z features detected by FI-MS based metabolomics analysis of serum samples with 20 m/z scan ranges in negative ionization mode. The measured ion intensity of SDS (sodium dodecyl sulfate; $[\text{M}-\text{H}]^- - 265.15$ Da; in red), TC (taurocholic acid; $[\text{M}-\text{H}]^- - 514.28$ Da; in red, dotted line), Caffeine ($[\text{M}-\text{H}]^+ - 195.09$ Da; in green), MRFA (Met-Arg-Phe-Ala peptide; $[\text{M}-\text{H}]^+ - 524.26$ Da; in green, dotted line) and NAD (β -Nicotinamide adenine dinucleotide; $[\text{M}-\text{H}]^+ - 664.12$ Da; in green, dashed line) when adding a minimal concentration of 10 μM of each compound. **b-k** The number of significant m/z features found (within the narrower scan range) when scanning for the 20 m/z scan range (in blue) and for the 24 m/z scan range (in red; the number in brackets in the sub titles correspond to one of the range borders), adding increasing concentrations of SDS (**b, d**), TC (**c, e**), Caffeine (**f, i**), MRFA (**g, j**) and NAD (**h, k**); black horizontal line represents the number of m/z features detected without adding these compounds.

Did you find that flow rate influences ion suppression (the infusion flow rate here is relatively high. Nanoflow has been shown to reduce ion suppression relative to $\mu\text{L}/\text{min}$ flow)?

Lowering the flow rate below $75 \mu\text{L min}^{-1}$ negatively affected the sensitivity of our FI-MS method. While Nanoflow technology indeed enable to lower the flow rate and reduce ion suppression in the ESI source, our results suggest that ion suppression in the ESI source is not the major factor that affects the number of detected m/z features; it is rather ion competition in the MS detection system, which would not be solved by lowering the flow rate. Using of nanoflow for high-throughput metabolomics has several potential disadvantages, including: (i) Lower flow rate could result in more time until sample is delivered to ESI source and then washed out; (ii) in some cases, Nanoflow results in unstable nESI spray (requiring prolong data acquisition to obtain a stable signal), as well as blocking of cheap, etc. (Southam et al., 2017); (iii) not fully automated, requiring replacement of nESI cheap, and expensive.

General: It is unclear what the final pipeline is for data acquisition in a biological study. Please define this. E.g. when using the final method for biological studies, do you acquire each sample multiple times or just once?

For the test data, samples were acquired 6 times in order to filter and retain high quality peaks. In practice when using this method, is there a need for technical replication of each biological sample to determine quality peaks from bad peaks. If so how does this increase the acquisition time for each sample? Can you clearly define the time length for analysing one biological sample (include any technical replicates and also instrument time between samples)?

The mass spectrometer acquisition time in our FI-MS method is 15 seconds in total (which suffices for scanning 8 ranges in negative mode and then 8 in positive; see Methods). The cycle time in our analysis is 30 seconds (i.e. time between the beginning of acquisition of two consecutive samples); achieved by optimizing flow rate gradient (as now elaborated upon in Supplementary Methods).

Samples were repeatedly injected 6 times as part of the method optimization process, evaluating signal to noise (SNR) versus blank of sample preparation (to eliminate bad peaks) and technical reproducibility in terms of RSD. An actual application of our method for analyzing a series of biological samples would not mandate repeated injections of each sample. Notably, whether repeated injections are needed or not depends on the specific application at hand and the desired analytical performance. Repeated injection

of a sample would obviously lower the uncertainty regarding the true abundance; i.e. for an ion with RSD of x , having n repeated injections would give a standard error (normalized by the mean) of $\frac{x}{\sqrt{n}}$.

In the revised version of the manuscript, we added an example of utilizing our FI-MS method to analyze 98 serum samples from healthy individuals, exploring the variation in metabolite abundance within the population: We analyzed 98 serum samples from healthy individuals using the developed FI-MS method. Samples were obtained from the Israeli Midgam Biobank (IRB: 0481-18-RMB). Quality Control (QC) samples were prepared by mixing 20 μL aliquots from each sample. A QC sample was injected every 5th serum sample and a blank sample every 10th serum sample. Significant m/z features were identified based on SNR and RSD of detected features in all QC samples (see Methods). To evaluate the importance of repeating the injection of samples, we injected every sample 4 times. We used the data to reproduce an analysis of gender prediction performed with LC-MS based metabolomics analysis of serum samples (Dunn et al., 2014), using our FI-MS method. Gender prediction was performed using a random forest model, consists of 100 decision trees, using the identified significant m/z features. Model accuracy was calculated based on out-of-bag observations. Considering a single injection of each serum sample, the measured intensities provided a gender prediction accuracy of 80% and 77%, in positive and negative ionization modes; comparable to those reported with the LC-MS analysis (78% and 85%, in positive and negative modes, respectively; slightly higher in positive and lower in negative modes) - though the FI-MS analysis was \sim 100-fold faster. The accuracy slightly increases with 2, 3, and 4, repeated injections of each sample (taking the median intensity from all replicates): 81% and 79% with 2 injections, 82% and 79% with 3 injections, and 83% and 81% with 4 injections (in positive and negative ionization modes, respectively; Supplementary Figure 8).

Supplementary Figure 8 Increase in accuracy of gender prediction across 98 samples of healthy individuals with 2, 3, and 4, repeated injections of each sample (taking the median intensity from all replicates) in negative (a) and positive (b) ionization modes.

Abstract, L17 – You state that LC-MS is not suitable for high throughput applications. This depends on how high-throughput is defined. 15 min LCMS metabolomics assays are fairly common and this can be used to analyse 1000+ samples which is normally considered high throughput. Also it is a trade-off, LCMS will give more structural information on compounds and separate isobaric compounds. Can you define what you consider high throughput and also discuss the trade-off between structural information and time (LCMS v FIMS).

We now clearly state that by high-throughput we mean applications requiring the analysis of thousands of samples. We now further emphasize the advantages of LC-MS in terms of identifying isobaric compounds, obtaining structural information, and higher sensitivity.

Abstract – insufficient method information is currently in the abstract, especially as this is a methods paper.

The Abstract was now revised to meet the 150-word limit in Nature Communications. We aimed to provide as much method information as possible.

Abstract, L26 – method is said to be 15 s long, is this correct? I'm not clear if the widows are 15s or the overall method. Also how long is instrument time between samples?

We now clearly indicate 15s is the method scan time (which includes 8 scans in both positive and 8 scans in negative ionization modes). The cycle time of this method is 30 seconds (time between the injection of consecutive samples).

Abstract, L26 –the term quantification could be misleading as metabolomics is considered semi-quantitative rather than fully quantitative.

We removed the term quantification from the Abstract. We now further utilize internal standards and standard addition method to quantify the absolute concentration of several amino acids and lipids, showing that metabolite concentrations inferred by FI-MS match those determined by LC-MS (Figure 3).

L37 – metabolomics should only be described as measuring ‘concentration’ if a chemical standard, internal standard and standard curve is also used.

We now refer to measuring ‘concentrations’ only when indeed utilizing chemical standards.

L43 – I wouldn’t describe NMR as high resolution. Also for MS high resolution only applies to some detectors.

We removed the ‘high-resolution’ term.

L48 – LCMS metabolomics assays for high throughput typically take 15 min with LC standard columns. These can be used for high throughput (depending on how high throughput is defined, see first point above).

L68-70 A method taking a few minutes would normally still be considered high throughput.

We now clearly state that by high-throughput we mean applications requiring the analysis of thousands of samples (e.g. useful for biomarker discovery and functional genomics screens).

L83 is your method 2 mins in length, i.e. 15 s x 8, or 15 s as suggested by the abstract?

We now clearly indicate 15 s is the method scan time (which includes 8 scans in both positive and 8 scans in negative ionization modes). The cycle time of this method is 30 seconds (time between the injection of consecutive samples).

L114 Section 2.5 doesn’t exist in the paper. Should this be ‘the methods section’?

Indeed, this was corrected.

L115-119 When considering detection sensitivity between 20 m/z and 24 m/z windows, did you only compare the features common to both windows (i.e. only those that fell within the 20 m/z windows)? This would be the fairest comparison.

Indeed, in order to perform the fairest comparison, we account for the features detected in the 20 m/z range only (lines 109).

L162-174 When counting features in the increasing window sizes, did you count the features in the entire window, or within a region common to all windows? The latter would be the fairest comparison.

Indeed, only features within the region common to all ranges were counted (lines 161-162).

L187-190 Again I'm not fully clear on total acquisition time for each method.

We now elaborate on how we performed FI-MS analysis with more than 8 scan ranges by injecting the same sample several times, while splitting the set of desired ranges to groups of 8 or less scans (see Methods).

L226 & L245 Define annotated. Were these putatively annotated based solely on accurate mass or was MS/MS also used. If it was accurate mass only, then they should be defined as putatively annotated.

We now refer to these as putative annotations based on accurate mass measurements.

L227-229 Was any extra annotation work done on these peaks other than accurate mass matching? Without this (e.g. MSMS annotations) you cannot be sure these actually are fatty acids, steroids and carbohydrates.

We explicitly relate to these as putative annotations and no additional work was done on annotating them (and no further analysis relies on these annotations).

L300 Did you check for cross contamination between samples? A good way to test would be running 2-3 blanks after a sample to see the degree of carry-over.

Possible cross contamination was checked during method development and now it is shown in Supplementary Methods (Optimization of gradient of flow rate) and Supplementary Figure 9. When computing the number of significant m/z features, we focused only on features with high SNR (> 4) compared to blank samples. Also, analyzing a series of blank samples ran after a biological sample, we do not detect any continuous drop in intensities due to carry-over (not shown).

L307-311 Why was there a gradient of flow? When you acquire data is the flow rate constant?

We now elaborate on how we optimized the parameters of the method in the Supplementary Information. Gradient of flow rate were used for fast sample delivery to ESI source and washing after

analysis to prevent possible cross contamination. The use of gradient of flow rate enabled to achieve a 30 seconds cycle time.

L318 On the Q Exactive HESI source 3 gasses exist: sheath, aux, sweep.

Is your method applicable for use on the HESI source? Which of these are the drying and nebuliser gasses?

Indeed, we used Thermo standard ESI source: In order to prevent miscomprehension all necessary corrections were made.

L342: 5ppm is a large mass tolerance when annotating based on only accurate mass. This will lead to many false positive identifications. The Q Exactive generally performs better than +/- 5ppm, have you assessed your system and considered decreasing the ppm tolerance window?

Indeed, the Q Exactive typically performs better than 5ppm. However, peak annotation is anyway not a main focus of our paper.

Figure 1 b,c & Fig 3 b, c: Add the name of the metabolite to the x axis to make reading easier.

Done.

*Points related to data processing and raw data:
L329: mzXML is a relative old data format / standard and is not maintained and further developed anymore. To make the implementation future proof the workflow should be compatible with mzML (more common format in Metabolomics and Proteomics) and mzXML formatted files.*

Other than the provided mzXML files, all raw mass-spectrometer files were uploaded to Zenodo repository (<https://doi.org/10.5281/zenodo.3581227>) and can be converted to any format of interest.

L332: The GitHub repository is not available via the link provided and therefore the implementation cannot be tested. Additionally, Matlab is not open source and a licence is required to test the code / implementation. This is a significant limitation and will limit the usability of the code/workflow provided. Therefore, consider other already existing tools written in open-source languages, such as R and python.

The GitHub repository is now available. Our entire code is already in Matlab, which indeed requires a license, but still considered a very common programming language for studies in this field.

L336: Define how a signal-to-noise ratio threshold of 4 was defined. What optimisation was used to define this threshold?

Various studies (Southam et al., 2017, Payne et al., 2009) choose SNR values in the range of 3-10 (higher SNR typically used to identify peaks that can be accurately quantified while lower SNR to identify peaks that are detectable). In any case, we compare our optimized scan range method with the existing uniform range mass stitching method using the same threshold.

It is unclear from the manuscript if the raw data has been submitted to one of the main metabolomics repositories, including Metabolights. I highly recommend making the raw data publicly available.

The raw data is publicly available in the Zenodo repository. (<https://doi.org/10.5281/zenodo.3581227>)

Reviewers' comments:

Reviewer #1 (Remarks to the Author):

The authors have adequately addressed my comments and have made a commendable effort to include what I believe will be both interesting and useful analyses for the community to improve on the state-of-the-art use of Orbitrap technology. Also the overall clarity has much improved, especially for potential non-expert readers.

I recommend this manuscript to be published in its current form.

A final note to the authors: The added benefit of using a standard reference material like NIST SRM1950 is that it is commercially available to the entire community, allowing an almost identical and long-term stable sample to be used across different labs as opposed to each lab or study making up its own test samples. As such results can be immediately compared, and the material can serve as a true benchmark in the community that new methods can be tested against - but this only works if labs actually use it.

Reviewer #2 (Remarks to the Author):

Using the extra compounds to check ion competition (fig S1):

- It is not clear why 2 plots per compound used. Can you clarify what the 3 numbers above the plot including the one in brackets correspond to?

- In S1(j) it appears that MRFA does induce quite strong ion suppression (though still not as much as the ion competition). Can you discuss this point in the paper and comment on its relevance to your method. Are there properties of MRFA which make it more susceptible to ion suppression? Do other chemically similar compounds also induce ion suppression?

Flow injection mass spectrometry methods section

- In the responses to comments, you say the scan time is 15 s including both pos and neg ion modes. In the methods section I can't see where you say both ion modes (pos and neg) are employed in a single injection. Do you employ the polarity switching on the QE? Or are 8 pos scans conducted, then 8 neg scans conducted? Can you clarify this in the methods.

Method validation against LCMS

There are a couple of places where the method FIMS is compared to LCMS to show that it gives similar results. There are some specific issues with these:

- Line 271-278. As you are trying to prove the outcome of your analysis and the previous LC-MS study are similar, you would expect similar fold changes in metabolites. However you appear to show the fold changes are higher than in the original study – why would the biology differ? I can't find any fold change data in either Supplementary data file 1 or Supplementary data file 2. Can you make the results available?

- Line 240-254. The FIMS method is used to predict gender from a large set of serum samples. This is proposed to show that it performs similar to LCMS, as this was shown previously in published data. I believe even the most crude metabolomics method (e.g. FIMS single mass range), would be able to determine male from female serum samples. So I don't believe this experiment shows that the FIMS method is equivalent to LCMS. A much better approach to validate the FIMS method against LCMS would be to carry out something as outlined in the 'Method sensitivity' section (below).

Method sensitivity

- Can you indicate how sensitive the method is in comparison to currently used metabolomics approaches? i.e. how many metabolites and lipids are consistently (and reproducibly) detected by your method and, for example, a standard 15 min LC-MS method and/or the current published FIMS stitching method you refer to in the paper (ref 17). This would be useful to the reader to assess a trade-off between time and sensitivity (sensitivity in-terms of actual useful metabolites or lipids detected).

Metabolite ID

- In your response, you say that peak annotation is not the main focus of the paper, so ID by MS/MS is not addressed. However, given that metabolite ID is the largest challenge in the field currently, it is important to know if peaks reported as putative IDs are likely to be actual metabolites/lipids.
- Nothing in the paper is identified by MS/MS, thus how can you be sure the peaks are indeed useful metabolites / lipids? Adding some MS/MS on lipids and expected metabolites and matching to online databases would add extra confidence in your method to the reader.
- Related to this, do you have a strategy for metabolite ID during experiments? You could also carry out a longer injection on a pooled sample and carry out DDA MS/MS on as many peaks as possible. This would be a useful addition to a study to provide some ID to be used further down the line during data analysis.
- Could you report the ppm mass error on the internal standards? This would help with putative ID. E.g. If the ppm error was small, you know you can set your ppm error on the putative metabolite search as small, thus reducing the number of false positives. You could also suggest the user infuses a known mix of compounds (possibly spiked into the biological matrix) as one of the samples in any given run. Post data collection, this could be assessed in-terms of mass error, thus allowing an appropriate mass error window to be set for putative searches on the rest of the dataset.

Other points

- The scan time is mentioned in the abstract as 15s. Can you indicate that this is pos and neg ion modes. Can you state the total duty time per sample (30 s I believe) – in both the abstract and towards the end of the introduction.
- Abstract: I'm not sure what the following statement means "a ~50% increase versus with the state-of-the-art techniques".
- Throughout the paper you refer to detected peaks as significantly detected peaks. To what does 'significantly' refer? Did you test these statistically? They are statistically changed relative to what? E.g . line 105 & 147

Tomer Shlomi, Associate Prof.
Faculty of Biology and Computer Science
Technion – Israel Institute of Technology
Haifa 32000, Israel
Tel: +972-4-829-4356
Fax: +972-4-829-3900

April 23, 2020

Dear reviewer,

We would like to thank you for the careful and thorough reading of this manuscript as well as the thoughtful comments and constructive suggestions, which have helped us improve the quality of the manuscript. The manuscript has been revised based on the remaining comments, as described below.

Reviewer #2 (Remarks to the Author):

Using the extra compounds to check ion competition (fig S1):

- It is not clear why 2 plots per compound used. Can you clarify what the 3 numbers above the plot including the one in brackets correspond to?

We evaluated the ion suppression and competition effects by injection a series of serum samples in which the ion flow was gradually induced by adding increasing concentrations of several compounds: 10 – 250 μM sodium dodecyl sulfate (SDS), taurocholic acid (TC) and Met-Arg-Phe-Ala peptide (MRFA); 10 – 1500 μM caffeine; 10 - 3000 μM β -Nicotinamide adenine dinucleotide (NAD). SDS and TC were used for investigation of ion suppression and competition effects in negative ionization mode (in both FI-MS based metabolomics and lipidomics analysis) and caffeine, MRFA and NAD in positive ionization mode (in FI-MS based metabolomics analysis; Supplementary Figures 1-2). For each of the above compounds, we performed a series of FI-MS runs while gradually increasing its concentration in the analyte, configuring a 20 m/z scan range which excludes this compound and an overlapping 24 m/z scan range that includes it. We repeated the experiment twice: Once, with both the 20 m/z and 24 m/z scan ranges starting at the m/z of the ion of the added compound minus 22 m/z units; and once, when the 20 m/z and 24 m/z ranges end at the m/z of the ion of the added compound m/z plus 22 m/z units (see Supplementary Figure 3). The effect of ion-suppression in the ESI was assessed based on the drop in the number of reproducibly detected m/z features in the 20 m/z scan range (which excludes the m/z signal of the added compound) as higher and higher concentration of the compound were added to the analyte. The total effect of ion competition in the detection system and ion suppression in ionization source was assessed based on the drop in the number of detected m/z features when configuring a 24 m/z range (that includes the m/z signal of the added compound; considering only m/z features within the smaller 20 m/z interval without the added compound). Following the reviewer's comment, we elaborated the explanation of these experiments in the Supplementary Methods; and added Supplementary Figure 3. The legends and titles were changed in Supplementary Figures 1 and 2 to specify the compound m/z and scan ranges in each experiment.

Supplementary Figure 3 An experimental scheme for investigating ion suppression and ion competition effects in FI-MS analysis: Gradually increasing the ion flow by adding increasing concentrations of different compounds to the analyte, while configuring the mass spectrometer to scan for two overlapping ranges that include or exclude the added compound; here, taurocholic acid (TC; [M-H]⁻ – 514.28 m/z) was added to metabolite extracts from serum samples, while a 20 m/z scan range, which excludes this compound and an overlapping 24 m/z scan range that includes it are scanned. We repeat the experiment twice: Once, limiting the mass-spec scan window to 20 m/z and 24 m/z ranges that start at the ion of added compound m/z minus 22 m/z (at m/z of 492; results shown in panel a); and second, in which the 20 m/z and 24 m/z ranges end at the ion of added compound m/z plus 22 m/z (at m/z of 536; results shown in panel b). **a,b** The number of reproducibly detected m/z features found (within the narrower scan range; y--axis) when scanning for the 20 m/z scan range (in blue) and for the 24 m/z scan range (in red), adding increasing concentrations of TC (x-axis).

- In S1(j) it appears that MRFA does induce quite strong ion suppression (though still not as much as the ion competition). Can you discuss this point in the paper and comment on its relevance to your method. Are there properties of MRFA which make it more susceptible to ion suppression? Do other chemically similar compounds also induce ion suppression?

We thank the reviewer for indicating the apparent stronger ion suppression induced by MRFA versus by other compounds. Following the reviewer's comment, we found that the strong ion suppression effect with MRFA inferred based on the scanned range of (527-547 m/z) that excludes MFA ($[M+H]^+$ 524.27 m/z) can be explained by the formation of a $[M+Na]^+$ adduct ion with 546.25 m/z , which is included within the scanned range, inducing an ion competition effect – leading to a major overestimation the ion suppression effect in this specific experiment with MRFA. This is clearly evident in Figure L1 below, showing that the intensity of the adduct ion increases proportionally with that of $[M+H]^+$ when adding increasing concentrations of MRFA.

Figure L1 Measured intensities of $[M+H]^+$ ion of MRFA (blue) and of $[M+Na]^+$ adduct ion (in red) when adding increasing concentrations of MRFA to metabolite extracts from a serum sample.

To overcome the ion competition effect induced by the MRFA adduct, we repeated this experiment while shifting the range to 526-546 m/z (and accordingly also the corresponding range that includes MRFA to 522-546 m/z). The revised analysis clearly shows that ion competition in the detection system rather than ion suppression in ESI is the prime reason for the reduced sensitivity of FI-MS – also when this is induced with MRFA (see updated Supplementary Figure 1j).

Supplementary Figure 1 Ion competition and ion suppression effects in the metabolomics analysis induced by adding a series of increasing concentrations of five compounds, evaluated via different scan ranges. **(a)** The distribution of \log_{10} intensities of significant m/z features detected by FI-MS based metabolomics analysis of serum samples with 20 m/z scan ranges in negative ionization mode. The

measured ion intensity of SDS (sodium dodecyl sulfate; $[M-H]^- - 265.1479\ m/z$; in red), TC (taurocholic acid; $[M-H]^- - 514.2844\ m/z$; in red, dotted line), Caffeine ($[M-H]^+ - 195.0877\ m/z$; in green), MRFA (Met-Arg-Phe-Ala peptide; $[M-H]^+ - 524.2650\ m/z$; in green, dotted line) and NAD (β -Nicotinamide adenine dinucleotide; $[M-H]^+ - 664.1164\ m/z$; in green, dashed line) when adding a minimal concentration of 10 μ M of each compound. **b-k** The number of significant m/z features found (within the narrower scan range) when scanning for the 20 m/z scan range (in blue) and for the 24 m/z scan range (in red), adding increasing concentrations of SDS (**b, d**), TC (**c, e**), Caffeine (**f, i**), MRFA (**g, j**) and NAD (**h, k**); black horizontal line represents the number of m/z features detected without adding these compounds.

Flow injection mass spectrometry methods section

- In the responses to comments, you say the scan time is 15 s including both pos and neg ion modes. In the methods section I can't see where you say both ion modes (pos and neg) are employed in a single injection. Do you employ the polarity switching on the QE? Or are 8 pos scans conducted, then 8 neg scans conducted? Can you clarify this in the methods.

Our method consists of eight scans in negative ionization mode followed by eight scans in positive ionization mode for the analysis of one sample within ~15 seconds of total scanning time. This is now clarified in the Methods section. An additional Supplementary figure was added to illustrate the specific scan ranges and polarization modes used throughout the ~15 seconds scanning time (Supplementary Figure 15).

Supplementary Figure 15 The configured scan ranges (y-axis) and ionization modes (positive in blue; and negative in red) throughout the ~15 seconds data acquisition period (x-axis) in our FI-MS method.

Method validation against LCMS

There are a couple of places where the method FIMS is compared to LCMS to show that it gives similar results. There are some specific issues with these:

- Line 271-278. As you are trying to prove the outcome of your analysis and the previous LC-MS study are similar, you would expect similar fold changes in metabolites. However you appear to show the fold changes are higher than in the original study – why would the biology differ? I can't find any fold change data in either Supplementary data file 1 or Supplementary data file 2. Can you make the results available?

Our FI-MS measurement of metabolite abundance changes between normoxia and hypoxia are in agreement with those of Frezza et al¹: We find that for a set of 31 metabolites whose concentration was reported in Frezza et al to increase under hypoxia based the LC-MS measurements, our FI-MS analysis also shows a significant increase in abundance in hypoxia (Wilcoxon rank-sum test, p -value $< 10^{-7}$; comparing the hypoxia-normoxia fold-change in the FI-MS measured abundance of these metabolites with those of other detectable metabolites). We do not find overall higher hypoxia-normoxia fold-changes in our FI-MS analysis versus fold-changes reported by Frezza et al; and rewrote the relevant text which might have implicated otherwise. Metabolite hypoxia-normoxia fold-changes measured in our study via FI-MS for the set of 31 metabolites (reported by Frezza et al to have a significantly increased concentration in hypoxia) are given in Supplementary Table 2. Notably, while our FI-MS based finding of metabolites whose concentration go up in hypoxia versus in normoxia significantly matches the LC-MS based measurements of Frezza et al., for specific metabolites the hypoxia-normoxia fold-change differs. For example, for glycerolphosphate we detect a 7.4 ± 0.6 fold-change versus a 1.9 ± 0.2 fold-change reported in Frezza et al; while for serine, we detect a 1.8 ± 0.4 fold-change versus a 2.8 ± 0.3 fold-change in Frezza et al. These differences may be due to slight experimental differences (e.g. with regard to the specific dialyzed fetal bovine serum used for cell culture). Applying LC-MS to measure the intensities of glycerolphosphate and serine in normoxia and hypoxia shows similar fold changes to those measured with our FI-MS method (6.4 ± 0.4 and 1.8 ± 0.2 for glycerolphosphate and serine, respectively).

- Line 240-254. The FIMS method is used to predict gender from a large set of serum samples. This is proposed to show that it performs similar to LCMS, as this was shown previously in published data. I believe even the most crude metabolomics method (e.g. FIMS single mass range), would be able to determine male from female serum samples. So I don't believe this experiment shows that the FIMS method is equivalent to LCMS. A much better approach to validate the FIMS method against LCMS would be to carry out something as outlined in the 'Method sensitivity' section (below).

Applying FI-MS to perform metabolomics analysis of 98 serum samples from healthy individuals enabled to reproduce previous results² that were obtained with LC-MS in terms of the distribution of inter-subject metabolite RSD %; and regarding specific classes of metabolites showing especially low/high inter-subject RSD % (see lines 245-259). Following the reviewer's comment, we now repeated the entire FI-MS analysis of 98 serum samples using: (i) Our optimized FI-MS method; (ii) Using uniform m/z ranges; and (iii) using a single m/z range. We injected every sample 4 times, resulting in a total of 1176 injections of serum samples. We repeated the gender prediction based on a similar random forest model as presented in the paper. Considering a single injection of each serum sample, the measured intensities provided a gender prediction accuracy of 80% and 82%, in positive and negative ionization modes; comparable to those reported with the LC-MS analysis (78% and 85%, in positive and negative modes, respectively). A somewhat lower accuracy of 74% and 76% was obtained with the uniform-range FI-MS; and 78% and 74% with a single-range FI-MS. The improved accuracy obtained with our optimized FI-MS method is further observed when performing multiple injections of each sample to lower noise and considering the median intensity per sample (Supplementary Figure 10).

Supplementary Figure 10 Accuracy of gender prediction across 98 serum samples of healthy individuals based on measurements performed with our optimized ranges FI-MS method (blue), uniform scan ranges (green), and using a single scan range (red), considering 1, 2, 3, and 4, repeated injections of each sample, in negative (a) and positive (b) ionization modes

Following the reviewer's comments, we further evaluated the performance of our method directly compared to LC-MS. Towards this end, we applied our optimized FI-MS method (as well as FI-MS with uniform ranges and with a single range) and untargeted LC-MS to analyze metabolite extracts from 10 cell lines: HeLa, Hek293, HepG2, MiaPaca2, HCT116, Panc-1, A549 and WM266-4, Jurkat and CCRF-CEM cells. We found a total of 815 m/z features that are identified by both LC-MS and FI-MS within at least 7 cell lines, in positive and negative ionization modes combined (utilizing MAVEN²⁸ to extract reproducible intense MS peaks with intensity > 15,000 in negative and positive ionization modes). Ion intensity measurements performed by FI-MS across cell lines are significantly correlated with those made by LC-MS for a total of 367 m/z features in negative and positive modes (FDR corrected Pearson $p < 0.05$; Supplementary Data File 3). For FI-MS with uniform ranges, a significant correlation with LC-MS measurements was obtained for only 216 m/z features; and for FI-MS with a single range for only 86 m/z features (Supplementary Figure 11). Furthermore, the correlations between LC-MS intensity measurements and those made via our optimized ranges FI-MS are significantly higher than those made with FI-MS with uniform ranges and with FI-MS with a single (Wilcoxon rank-sum test p -value $< 10^{-3}$ and $< 10^{-7}$, respectively). For example, for uridine diphosphate *N*-acetylglucosamine, the correlation between measurements performed with our optimized ranges FI-MS (negative ionization mode) and LC-MS ($r = 0.95$, FDR corrected p -value $< 10^{-6}$) is markedly higher than that for uniform ranges FI-MS ($r = 0.87$, FDR corrected p -value < 0.05) and single range FI-MS ($r = 0.78$, FDR corrected p -value > 0.05).

Supplementary Figure 11 Number of m/z features (y-axis) detected by our optimized ranges FI-MS method (blue), uniform ranges (spectral-stitching) FI-MS (green), and single range FI-MS (red) across a panel of 10 cell lines (HeLa, Hek293, HepG2, MiaPaca2, HCT116, Panc-1, A549 and WM266-4, Jurkat and CCRF-CEM) whose intensity profile across cell lines is significantly correlated with corresponding LC-MS measurements (FDR corrected Pearson $p < 0.05$), in negative and positive modes.

Method sensitivity

- Can you indicate how sensitive the method is in comparison to currently used metabolomics approaches? i.e. how many metabolites and lipids are consistently (and reproducibly) detected by your method and, for example, a standard 15 min LC-MS method and/or the current published FIMS stitching method you refer to in the paper (ref 17). This would be useful to the reader to assess a trade-off between time and sensitivity (sensitivity in-terms of actual useful metabolites or lipids detected).

Figure 2e shows a comparison of our optimized ranges FI-MS method with the current published FI-MS ion-stitching method (with uniform scan ranges; considering the same number of 8 scan ranges as in our optimized ranges FI-MS), and also compared to an even more standard FI-MS with a single scan range: We find a significant ~50% increase in the number of reproducibly detected m/z features in metabolite extracts from serum samples compared to the published FI-MS method with uniform ranges and a significant ~390% increase compared to standard FI-MS with a single scan range (two-sample t-test p-value $< 10^{-7}$ and p-value $< 10^{-10}$, respectively; Figure 2e; Methods).

Following the reviewer's comment, we now also compare the FI-MS methods not only in terms of the number of reproducibly detected m/z features, but also in terms of the expected number of distinct metabolites identified by each method (considering that multiple ions may be associated with a single metabolite, due to adducts, fragments, and natural isotopes). This is performed in two ways:

- (i) **Based on the number of detected m/z features with a putative annotation:** We obtained putative annotations for m/z features detected with each of the three FI-MS methods, based on the high-accuracy Orbitrap MS measurements, compared with HMDB and LMSD (further MS/MS annotation was also performed, please see reply below in 'Metabolite ID' section). We find that the number of putative annotations for reproducibly detected m/z features with our optimized ranges FI-MS method is ~2-fold higher than for the set of m/z features detected with uniform ranges FI-MS; and ~3.5-fold higher than for the set of m/z features detected with FI-MS with a single range (Supplementary Figure 8).

Supplementary Figure 8 The number of putatively annotated m/z features for measurements performed with our optimized ranges FI-MS method (blue), uniform ranges FI-MS (green), and single range FI-MS (red). * $P < 0.01$ by two-sample t-test. Data are mean \pm SD, $n = 5$ independent repetitions of the FI-MS analysis.

- (ii) **Based on m/z clustering of multiple ions associated with the same metabolite (due to adducts, fragments, and natural isotopes):** Towards this end, we adapted a method used to group m/z peaks in standard LC-MS analysis⁴ to clustering of reproducibly detected m/z features in FI-MS analysis. Specifically, while with LC-MS data, m/z peaks are clustered based on a correlation between peak shapes (i.e. time-dependent intensity measurement throughout the chromatographic separation), with FI-MS, m/z features are clustered based on a high correlation between the measured intensity across a series of analyzed samples (see Supplementary Methods). Towards this end, we utilized our FI-MS based metabolomics data for 98 serum samples (whose measurement is described above). Using a stringent criterion of above 0.95 Pearson correlation between m/z features to be included within a single cluster, we detect a total of ~ 3900 distinct m/z feature clusters for measurements performed with our optimized ranges FI-MS; compared to ~ 1200 m/z clusters obtained by the uniform ranges FI-MS, and only ~ 250 clusters with a single range FI-MS. The marked increase in the number of m/z feature clusters obtained by our optimized ranges FI-MS method is clearly evident across a wide range of thresholds on the minimal correlation between m/z features to be clustered (Supplementary Figure 12).

Supplementary Figure 12: The number of m/z clusters identified with metabolomics analysis of 98 serum samples (y-axis), performed with our optimized ranges FI-MS method (blue), uniform ranges FI-MS (green), and single range FI-MS (red); considering a range of thresholds on the minimal correlation between the measured intensity of m/z features across samples that are clustered together (x-axis).

Overall, we have shown that our optimized ranges FI-MS method outperforms the previously published uniform ranges (mass-stitching) FI-MS and single range FI-MS methods, providing a higher number of reproducibly detectable m/z features, estimated number of distinctly detectable metabolites via m/z feature clustering, and metabolites with a putative annotation. Furthermore, ion abundance measurements performed with our optimized ranges FI-MS show higher correlations with LC-MS based measurements, compared to measurements performed via the published uniform ranges FI-MS and single range FI-MS methods.

Metabolite ID

- In your response, you say that peak annotation is not the main focus of the paper, so ID by MS/MS is not addressed. However, given that metabolite ID is the largest challenge in the field currently, it is important to know if peaks reported as putative IDs are likely to be actual metabolites/lipids.
- Nothing in the paper is identified by MS/MS, thus how can you be sure the peaks are indeed useful metabolites / lipids? Adding some MS/MS on lipids and expected metabolites and matching to online databases would add extra confidence in your method to the reader.

We evaluated the analytical performance of our optimized ranges FI-MS method versus uniform ranges FI-MS in terms of the number of reproducibly detectable m/z features – i.e. metabolites/lipids which are reproducibly detected across multiple injections with an intensity several fold higher than in blank samples. We also suggested putative annotations for identified m/z peaks based on the high mass-accuracy measurements compared to public databases (level 2 annotation, as defined by the MSI⁵), in accordance with previous studies utilizing FI-MS^{6,7}. We certainly agree with the reviewer that metabolite identification is a major challenge in the field, and hence while the focus of our paper is not annotation (but rather establishing a methodology for both rapid and high sensitivity FI-MS metabolomics), we now also demonstrate usage of MS/MS to refine the annotation of identified m/z peaks – see below.

- Related to this, do you have a strategy for metabolite ID during experiments? You could also carry out a longer injection on a pooled sample and carry out DDA MS/MS on as many peaks as possible. This would be a useful addition to a study to provide some ID to be used further down the line during data analysis.

We now performed further FI-MS analysis with DDA MS/MS mode on serum samples, splitting the m/z interval from 70 to 990 m/z to 13 ranges of 80 m/z width. For each range, we performed a separate FI-MS analysis, once in negative and once in positive ionization modes (using the same injection volume, flow gradients, and MS ionization source parameters as described in the paper for our optimized ranges FI-MS method). Carrying out a longer injection with ESI by substantially slowing down the flow rate would lower the sensitivity of the method and damage reproducibility.

MS² data was obtained for a total of 36 and 47 precursor ions in negative and positive ionization modes respectively (with at least a single ion in the MS² spectra with an intensity > 10,000). Annotation of precursor ions was performed based on a match with publicly available METLIN MS² spectra (using the

online fragment similarity search tool) for polar compounds, and LMSD bulk Structure Search and LMSD Product ion calculation tool for prediction of MS/MS fragments for non-polar compounds. Utilizing METLIN MS² spectra, we annotated 22 polar precursor ions, 20 with a unique hit (with a match in 1 to 4 collisional fragments) and 2 with two possible hits due to isomers. Using LMSD, we annotated another 61 non-polar compounds. For many of the latter, FI-MS enables annotation of lipid class, total length of all acyl chains, and total number of double bonds, while more specific annotation would require an extra level of separation beyond m/z (Supplementary Data File 4).

We now relate to the important challenge of annotation for m/z features identified with FI-MS in the Discussion, demonstrating how MS/MS could be used for that. We further note that FI-MS with parallel reaction mode (PRM) could be systematically utilized to acquire MS/MS spectra for numerous other m/z features of interest via a series of rapid FI-MS.

- Could you report the ppm mass error on the internal standards? This would help with putative ID. E.g. If the ppm error was small, you know you can set your ppm error on the putative metabolite search as small, thus reducing the number of false positives. You could also suggest the user infuses a known mix of compounds (possibly spiked into the biological matrix) as one of the samples in any given run. Post data collection, this could be assessed in-terms of mass error, thus allowing an appropriate mass error window to be set for putative searches on the rest of the dataset.

Following the reviewer's comments, we repeated the analysis of 98 serum samples using our optimized ranges FI-MS method, adding a mix of internal standards to the extraction solution, enabling to infer MS mass accuracy: μM of ¹³C₁₁-tryptophan (MW – 215 Da), 0.5 μM of puromycin dihydrochloride (MW – 471 Da) and kiton red 620 (MW – 580 Da). We find mass accuracy of up to 5ppm, which was used for m/z feature annotation (Supplementary Table 4). We now specify in the Methods how mass accuracy was determined based on internal standards (see lines 455-458).

Other points

- The scan time is mentioned in the abstract as 15s. Can you indicate that this is pos and neg ion modes. Can you state the total duty time per sample (30 s I believe) – in both the abstract and towards the end of the introduction.

Please see our reply above in “Flow injection mass spectrometry methods section”. The total duty time (time between two consecutive injections) per sample is 30s and the scanning time including both positive and negative ionization is 15s. This is now clarified in the Methods section, illustrated in a new figure (Supplementary Figure 15), and the total duty time is now also specified in the Abstract.

- Abstract: I’m not sure what the following statement means “a ~50% increase versus with the state-of-the-art techniques”.

Throughout the manuscript we compared our FI-MS method to the published uniform ranges FI-MS (also referred to as “mass-stitching”), and as shown in Figure 2e, f our optimized FI-MS results in a ~50% percent increase in the number of reproducibly detected m/z features. This is now clarified in the Abstract.

- Throughout the paper you refer to detected peaks as significantly detected peaks. To what does ‘significantly’ refer? Did you test these statistically? They are statistically changed relative to what? E.g . line 105 & 147

Significantly detected m/z features were defined based on the following criteria (in accordance with previous studies⁸⁻¹⁰): (1) Observed in 90% of the biological sample injections; (2) The median intensity of the biological sample injections is above 1000 units; (3) Signal-to-noise ratio above 4; i.e. the median intensity in the biological sample injections divided by the maximal intensity of the blank injections. (4) Relative standard deviation (RSD) across the biological sample injections lower than 30%.

Following the reviewer’s comment, we rephrased the term “significantly detected m/z features” to “reproducibly detected m/z features” throughout the manuscript.

Sincerely yours,

Tomer Shlomi

References

1. Frezza, C. *et al.* Metabolic profiling of hypoxic cells revealed a catabolic signature required for cell survival. *PLoS One* **6**, (2011).
2. Dunn, W. B. *et al.* Molecular phenotyping of a UK population: defining the human serum metabolome. *Metabolomics* **11**, 9–26 (2014).
3. Melamud, E., Vastag, L. & Rabinowitz, J. D. Metabolomic analysis and visualization engine for LC - MS data. *Anal. Chem.* **82**, 9818–9826 (2010).
4. Kuhl, C., Tautenhahn, R., Böttcher, C., Larson, T. R. & Neumann, S. CAMERA: An integrated strategy for compound spectra extraction and annotation of liquid chromatography/mass spectrometry data sets. *Anal. Chem.* **84**, 283–289 (2012).
5. Sumner, L. W. *et al.* Proposed minimum reporting standards for chemical analysis Chemical Analysis Working Group (CAWG) Metabolomics Standards Initiative (MSI). *Metabolomics* **3**, 211–221 (2007).
6. Southam, A. D., Weber, R. J. M., Engel, J., Jones, M. R. & Viant, M. R. A complete workflow for high-resolution spectral-stitching nanoelectrospray direct-infusion mass-spectrometry-based metabolomics and lipidomics. *Nat. Protoc.* **12**, 255–273 (2017).
7. Ortmayr, K., Dubuis, S. & Zampieri, M. Metabolic profiling of cancer cells reveals genome-wide crosstalk between transcriptional regulators and metabolism. *Nat. Commun.* **10**, 1841 (2019).
8. Dunn, W. B. *et al.* Procedures for large-scale metabolic profiling of serum and plasma using gas chromatography and liquid chromatography coupled to mass spectrometry. *Nat. Protoc.* **6**, 1060–1083 (2011).
9. Broadhurst, D. *et al.* Guidelines and considerations for the use of system suitability and quality control samples in mass spectrometry assays applied in untargeted clinical metabolomic studies. *Metabolomics* **14**, 1–17 (2018).
10. Payne, T. G., Southam, A. D., Arvanitis, T. N. & Viant, M. R. A Signal Filtering Method for Improved Quantification and Noise Discrimination in Fourier Transform Ion Cyclotron Resonance Mass Spectrometry-Based Metabolomics Data. *J. Am. Soc. Mass Spectrom.* **20**, 1087–1095 (2009).

REVIEWERS' COMMENTS:

Reviewer #2 (Remarks to the Author):

Thank you for the thorough responses to my questions. I'm happy with the responses from the authors and I recommend accept with a couple of minor corrections below.

(i) when referring to a metabolite in the text, can you clearly indicate the level of annotation (i.e. the international MSI level).

(ii) when comparing FIMS data to previous hypoxia study (Supplementary Table 3), can you indicate if metabolites that were decreased in hypoxia in the original study are also decreased in your data. So far the data just compares metabolites that are increased in hypoxia.

May 24, 2020

The manuscript has been revised based on the remaining comments of reviewer #2, as described below, and following the editorial comments as described in the cover letter.

Reviewer #2 (Remarks to the Author):

Thank you for the thorough responses to my questions. I'm happy with the responses from the authors and I recommend accept with a couple of minor corrections below.

(i) when referring to a metabolite in the text, can you clearly indicate the level of annotation (i.e. the international MSI level).

This is now indicated in the manuscript.

(ii) when comparing FIMS data to previous hypoxia study (Supplementary Table 3), can you indicate if metabolites that were decreased in hypoxia in the original study are also decreased in your data. So far the data just compares metabolites that are increased in hypoxia.

The previous hypoxia study using LC-MS reported only a single metabolite that decreased in hypoxia and this was not reproducible in our LC-MS analysis; hence this was not referred to in the manuscript.

Sincerely yours,

Tomer Shlomi